# Offshore wind and wave energy can reduce total installed capacity required in zero-emissions grids

Natalia Gonzalez [1,2] ✉, Paul Serna-Torre [1,2], Pedro A. Sánchez-Pérez[3], Ryan Davidson[4], Bryan Murray [5], Martin Staadecker [1,6], Julia Szinai [7], Rachel Wei[8], Daniel M. Kammen [9], Deborah A. Sunter [10] & Patricia Hidalgo-Gonzalez [1,2] ✉

As the world races to decarbonize power systems to mitigate climate change, the body of research analyzing paths to zero emissions electricity grids has substantially grown. Although studies typically include commercially available technologies, few of them consider offshore wind and wave energy as contenders in future zero-emissions grids. Here, we model with high geographic resolution both offshore wind and wave energy as independent technologies with the possibility of collocation in a power system capacity expansion model of the Western Interconnection with zero emissions by 2050. In this work, we identify cost targets for offshore wind and wave energy to become cost effective, calculate a 17% reduction in total installed capacity by 2050 when offshore wind and wave energy are fully deployed, and show how curtailment, generation, and transmission change as offshore wind and wave energy deployment increase.

Power systems around the world are changing drastically as countries race to decarbonize in an effort to curb climate change. Traditional fossil fuel generators must be replaced with a diverse mix of renewable energy and other distributed energy resources to reduce carbon emissions while also maintaining grid stability and serving society's increasing demand for power. However, it remains as an open research question what the optimal mix of technologies is for decarbonizing power sectors in different regions across the globe.

Several studies[1–6] utilize different types of capacity expansion models to understand what future cost-optimal low-carbon electricity mixes may look like for the U.S. A few of these studies[1–3] provide analyses that seek to understand the role specific technologies may have in a future cost-optimal, low-carbon U.S. grid. For

example, the authors of ref. 1 investigate the impact that integrating bioenergy with carbon capture and sequestration (BECCS) with significant renewable deployment may have on enabling a carbon-negative power system in western North America by 2050. Similarly, the authors of ref. 2 investigate the role of firm low-carbon resources, such as nuclear, reservoir hydropower, geothermal, bioenergy, and natural gas, with carbon capture in reducing the cost of a decarbonized power grid. The authors of ref. 3 investigate how concentrated solar power with thermal energy storage (CSP+TES) competes with short-duration storage. Although refs. 1–4 do consider some promising technologies, they omit other emerging technologies that may be available in the future for commercial deployment, such as offshore wind and wave energy.

[1]Mechanical and Aerospace Engineering, University of California San Diego, 9500 Gilman Dr., La Jolla 92093 CA, USA. [2]Center for Energy Research, University of California San Diego, 9500 Gilman Dr., La Jolla 92093 CA, USA. [3]School of Engineering, University of California Merced, 5200 Lake Rd, Merced 95340 CA, USA. [4]CalWave, 1595 Portland Ave, Berkeley 94707 CA, USA. [5]University of Oviedo, C. San Francisco, Oviedo, 3, 33003 Asturias, Spain. [6]Division of Engineering Science, University of Toronto, 27 King's College Cir, Toronto, ON M5S 1A1, Canada. [7]Lawrence Berkeley National Laboratory, 1 Cyclotron Rd, Berkeley 94720 CA, USA. [8]Computer Science and Engineering, University of California San Diego, 9500 Gilman Dr., La Jolla 92093 CA, USA. [9]Renewable and Appropriate Energy Laboratory, Energy Resources Group, University of California Berkeley, 339 Giannini Hall, Berkeley 94720 CA, USA. [10]Civil and Environmental Engineering, Tufts University, 200 College Ave, Medford 02155 MA, USA. ✉e-mail: n7gonzalez@ucsd.edu; phidalgogonzalez@ucsd.edu

The study presented in this manuscript focuses on the Western Interconnection of the U.S. Many states[7–11] that constitute the Western Interconnection have pledged to achieve clean energy goals by 2030–2050. In light of these ambitious targets, wave energy and offshore wind energy, two co-existing abundant resources on the West Coast of the U.S., arise as clean energy sources to be considered in the transition towards carbon-free generation portfolios of the Western Interconnection.

The resource potential for both offshore wind and wave energy is tremendous along the West Coast of the U.S. (California, Oregon, Washington)[12–14]. In the U.S. West Coast, the offshore wind energy potential is 800 TWh/yr[14] and the wave energy potential is 240 TWh/yr[15], together demonstrating ~1.2 times the 2021 annual electricity demand in the Western Interconnection[16]. Despite this potential, there are no offshore wind turbines off the coast of the Western U.S. There are only two commercial offshore wind farms off the coast of the Eastern U.S. (a 5-turbine farm and a 12-turbine farm), as well as two demonstration offshore wind turbines off the coast of Virginia. There are no commercially operating wave energy farms on either coast[17].

Sharing the same hostile marine environment, similar obstacles have prevented cost parity with other technologies in the grid and hindered the growth of both of these technologies. Some of these obstacles include high maintenance costs due to intense ocean conditions, environmental disruption concerns, permitting challenges due to marine zoning laws, and visual impact concerns for near-shore deployment[18]. Current offshore wind and wave energy technologies are not cost-competitive with fossil fuel technologies, onshore wind energy, or solar energy either historically or with projections into 2030[17,19–21]. However, as we move towards a decarbonized energy future, we would benefit by considering a diverse portfolio of renewable energy sources. The technical benefits of integrating offshore wind and wave energy, coupled with cost reductions that would take place from deploying them, may make them contenders in the future.

Wave energy has several attributes that are advantageous. For example, wave energy is predictable up to 3 days in advance and is more consistent than most other renewable alternatives[22]. In some regions, when compared to wind energy, wave energy has less visual impact and higher energy density, as well as more continuous and predictable power output[23]. Furthermore, integrating wave energy with other renewable technologies can be complementary in nature. For example, ref. 23 studies how wave energy has decoupled weather patterns to solar, depending on the local conditions. Hence, coupling these technologies can have balancing effects. For the U.S. West Coast, wave energy is expected to have approximately four times more energy availability during winter months than summer months, as demonstrated by the PacWave test site off the coast of Newport, OR[24]. Deploying wave energy on offshore wind farms could also have similar power output smoothing effects, especially in areas with low correlations between wind and wave conditions or with a lag between the two power sources, as the authors of ref. 23 discuss. Additionally, coupling wave energy with offshore wind could provide enhanced energy yield and better predictability. One study showed that combined wind-wave farms in California would have fewer than 100 h of no power output per year, compared to >1000 h for offshore wind or more than 200 h for wave farms alone[25]. Other advantages that combined wind-wave farms have over traditional offshore wind farms include more efficient utilization of offshore site areas, shared project development costs between the two technologies, shared underwater transmission costs, shared substructure foundations, and reduced environmental impacts[26,27].

Offshore wind also has several technical and social benefits. The most attractive attribute of offshore wind is the higher capacity factors it yields compared to land-based wind[28]. Additionally, offshore wind tends to have a higher public acceptance than land-based wind and other land-based renewable technologies because the public does not experience significant visual impacts, noise production, or shadow casting from wind turbines if they are placed sufficiently far from the shore[28].

Few studies consider the role and system-wide impacts that wave and offshore wind technologies may have on the grid when they are deployed. The work in ref. 23 analyzes the value and effects that wave energy combined with offshore wind energy can have on southern Sweden's electricity grid. However, since the work conducts the analysis with a production cost model, it does not consider the investment costs of generating units or transmission infrastructure. The authors of ref. 29 analyze the effects of wave energy on the Southwest United Kingdom grid, but it focuses only on the effects of considering multiple wave energy sites on the quality of power output in terms of reduced intermittency of supply and step changes in generated power. The scope of ref. 30 is limited to studying the impact of one particular wave energy test site off the coast of Oregon, U.S., on the local grid in terms of steady-state, dynamic, and transient characteristics.

Several studies[31–34] analyze the impacts of integrating specifically offshore wind farms into the power grid, but they are limited to analyzing only the effects caused by offshore wind integration on voltage and frequency stability and wholesale prices in the electricity market, or their results do not stem from an optimization framework or the use of a real grid (stylized small power networks). A recent study analyzes the role of offshore wind in decarbonizing the U.S. using a capacity expansion model and various scenarios centered around policy and demand, technology cost and availability, transmission, and sitting constraints for various generation options[35]. Although this study is thorough in reviewing factors that shape offshore wind deployment and how those factors affect the role that offshore wind plays in achieving various levels of grid decarbonization, it does not include wave energy or explore the interplay between offshore wind and wave energy with the potential for collocation.

Despite the large body of work analyzing the potential electricity generation mixes for a future decarbonized U.S. grid and the several studies investigating certain impacts of integrating offshore wind and/or wave energy, the literature falls short of including both fixed-bottom and floating offshore wind and wave energy in the mix of candidate renewable technologies and understanding the technical implications of their relative deployment. To address this gap, this paper investigates the system-wide impacts of integrating various amounts of offshore wind (both fixed-bottom and floating) and wave energy into a carbon-free electricity mix using. For this, we use a least-cost capacity planning model with high spatial resolution, detailed power systems modeling, and a wide variety of candidate technologies.

We model the Western Interconnection with a 2050 zero-emissions future using SWITCH[36], a long-term capacity expansion model that has been used in numerous studies of low- or zero-emissions electricity grids[1,37–39]. Our model also contains 7000+ candidate plants that the model may choose to build. These plants are distributed across 50 load zones that cover the Western Electricity Coordinating Council (WECC) and are connected by 126 aggregated transmission lines. Additionally, the model simultaneously optimizes investment and dispatch decisions to minimize the total system cost and meet each load zone's power demand while considering the transmission network. Dispatch decisions are made at consecutive four-hour intervals for two representative days per month for investment periods 2020, 2030, 2040, and 2050. The year 2050 is when WECC-wide carbon emissions from electricity generation are required to reach zero in all scenarios. Since we seek to understand how integrating offshore wind and wave energy affects the cost-optimal zero-emissions system, the results presented in this paper focus on the year 2050. Results for investment periods 2020–2040 are presented in section 3 of the Supplementary Information.

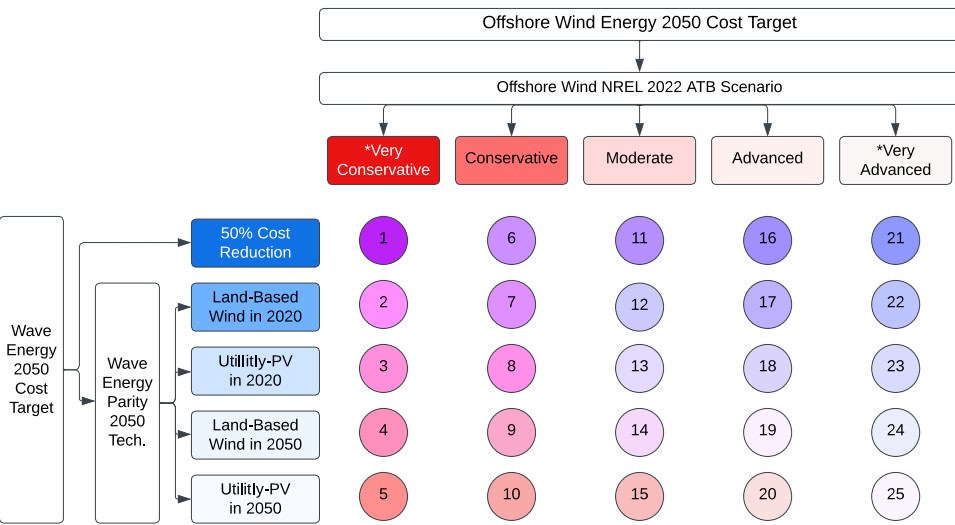

**Fig. 1 | Scenario numbering according to relative cost targets of offshore wind and wave energy.** The row labels describe which technology wave energy is assumed to reach cost parity by 2050 in each of the 25 scenarios designed for this study. Note that a 50% cost reduction for wave energy corresponds to $1732.50/kW overnight cost and $52.70/kW-yr O&m cost for wave energy in 2050. The column labels describe which offshore wind NREL 2022 ATB scenario is assumed for the cost of offshore wind energy in each of the 25 scenarios designed for this study. *Derived by offsetting an NREL 2022 ATB projection.

We design five different cost targets for wave energy. The most conservative cost target corresponds to a 50% cost reduction by the year 2050. The other four cost targets, in order from more conservative to more optimistic, are such that the 2050 overnight and operation and maintenance (O&M) wave energy costs reach parity with the National Renewable Energy (NREL) 2022 Annual Technology Baseline (ATB)[40]: (2) land-based wind 2020 costs, (3) land-based wind projected 2050 costs, (4) utility-scale photovoltaic (PV) energy 2020 costs, and (5) utility-scale PV energy projected 2050 costs, assuming a linear projection between the wave energy 2020 and 2050 costs. Similarly, we design five offshore wind overnight and O&M cost targets. The three core cost targets follow the NREL 2022 ATB conservative, moderate, and advanced cost projections for fixed and floating offshore wind energy. We also designed an additional very conservative and very advanced offshore wind cost target.

We combine the five wave energy cost targets and five offshore wind cost targets into 25 (5 × 5) scenarios that pair each wave energy cost target with each offshore wind energy cost target to evaluate how the relative cost assumptions affect the optimal zero-emissions 2050 energy mix. Figure 1 summarizes the scenario number assigned to each combination of offshore wind and wave energy cost targets.

The simultaneous decrease in costs of offshore wind and wave energy technologies captured by the scenarios reflects a positive feedback loop in research and deployment for offshore energy technologies. If more offshore wind gets deployed over time, economies of scale and built infrastructure could positively affect further cost declines for offshore wind, and at the same time, it could lower entry barriers for other offshore technologies, such as wave energy. On the other hand, the scenarios that represent offshore wind and/or wave energy becoming significantly less expensive by 2050 while the other technology remains expensive capture the possibility that one technology may be substantially more invested in as we transition to renewable energy while the other is left behind.

The purpose of these scenarios is not to predict cost targets or likely trends for offshore wind and wave energy overnight and O&M costs in coming years, but rather they serve to answer What if? questions related to offshore wind and wave energy becoming cost-competitive with other renewable resources to varying degrees.

The main contributions of this work are the following: (1) modeling offshore wind and wave energy as independent technologies with the possibility of collocation in a power system capacity expansion model of the Western Interconnection, (2) identifying, cost targets for offshore wind and wave energy to become cost-effective in a zero-emissions grid, (3) observing a 17% of reduction in total installed capacity by 2050 when offshore wind and wave energy are fully deployed, and (4) quantifying how lower wave energy cost targets result in lower total transmission expansion, and on the other hand, lower offshore wind cost targets result in higher transmission expansion. We find that if wave energy reaches cost parity with land-based wind by 2050 and offshore wind energy aligns with the advanced offshore wind NREL 2022 ATB scenario, then wave energy and offshore wind energy can reach about 6% and 9% deployment in a cost-optimal zero-emissions Western Interconnection, respectively.

## Results

### Total installed capacity of the zero-carbon grid decreases

In general, as offshore wind and wave energy 2050 cost targets decrease, and consequently their deployment in the grid in 2050 increases, the total installed zero-emissions generation capacity in the Western Interconnection decreases (Fig. 2a). The overall installed capacity decreases by a maximum of 133 GW between scenario 1 (most expensive offshore wind and wave energy cost targets) and scenario 24 (very advanced offshore wind energy cost target and wave energy cost parity with land-based wind in 2050). This corresponds to a 17% decrease in total installed capacity in the grid, which is mostly driven by decreased cost targets of offshore wind energy, and, thus, increased deployment of offshore wind energy, as seen in Fig. 2a. When wave energy cost target decreases from the most conservative to the most optimistic wave energy cost target, we see a maximum decrease in total installed capacity of 3%.

The significant reduction in installed capacity across the scenarios implies that offshore wind and wave energy may play a key role in limiting the overbuilding of the grid to ensure demands are met in the future 2050 zero-emissions grid, even if they make up a relatively small portion of the total electricity mix. One of the factors that partially contributes to this reduction in total installed capacity is related to installed solar energy capacity: as more offshore wind and wave energy are deployed across the scenarios, the amount of solar capacity that needs to be installed in the zero-emissions Western Interconnection in 2050 decreases, as seen in Fig. 2b, although solar consistently remains the dominant source of energy for electricity generation. We observe a difference of 132 GW of solar installed between scenarios 1 and 25,

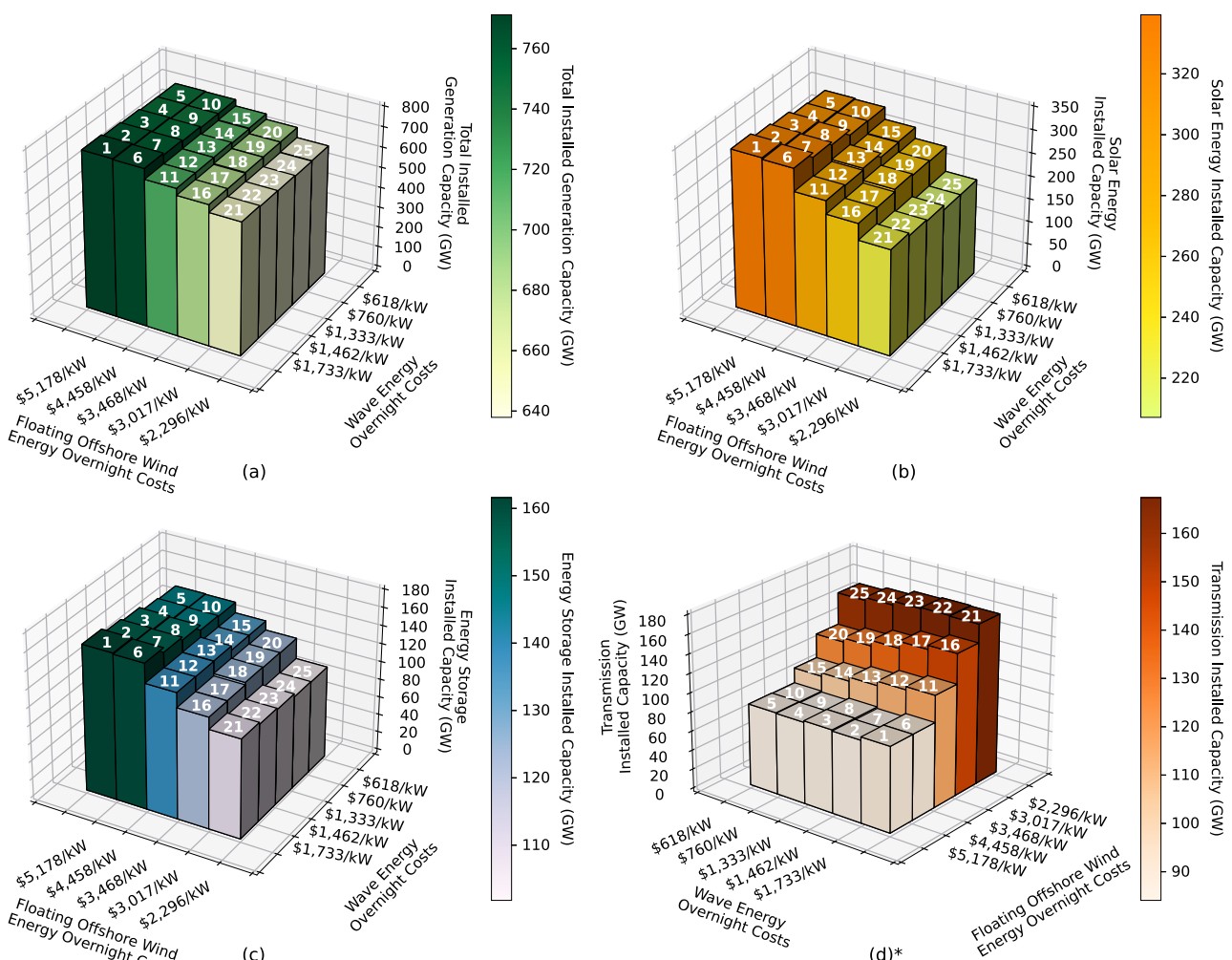

**Fig. 2 | Infrastructure results.** Scenario numbers are displayed at the top of each bar. **a** Total 2050 installed zero-emissions generation capacity (GW) in the Western Interconnection in each scenario. Total installed capacity decreases with decreasing offshore wind energy cost targets and mostly decreases with decreasing wave energy cost targets until offshore wind energy costs decline beyond the NREL ATB moderate scenario. **b** 2050 solar energy installed capacity (GW) in each scenario. Solar energy installed capacity decreases with decreasing offshore wind and wave energy cost targets. **c** 2050 energy storage installed capacity (GW) in each scenario. Energy storage installed capacity decreases with decreasing offshore wind and wave energy cost targets. **d** 2050 total land-based transmission capacity (GW) in the Western Interconnection for each scenario. Transmission capacity decreases with decreasing wave energy cost targets and increases with decreasing offshore wind energy cost targets. Note that the *x* and *y* axes are flipped in plot (**d**). This is done so that the trend is fully visible. Source data are provided as a Source Data file.

which is a ~39% decrease. As a reminder, scenario 25 assumes wave energy reaches cost parity with utility PV energy in 2050, and offshore wind energy costs align with the very advanced NREL 2022 ATB scenario (which is an offset of NREL's advanced scenario). Hence, if offshore wind and wave energy costs decline dramatically in the coming decades, these technologies have the potential to significantly reduce how much installed solar energy is required in the future zero-emissions grid.

Less deployment of solar energy consequently reduces mid-day over-generation and hence reduces reliance on energy storage. We observe that lower offshore wind and wave energy costs lead to lower storage capacity installed in the Western Interconnection in 2050. This effect is most dramatically seen with more rapidly declining offshore wind costs (Fig. 2c). We observe a maximum difference of 60 GW of storage installed (37% decrease) across scenarios. This decrease corresponds to a decline from 44% to 32% in terms of the share of total installed capacity made up by energy storage.

While solar energy remains the dominant technology across all scenarios, the reduction of solar energy and storage charging peaks in the grid achieved by increased deployment of offshore wind and wave energy may be beneficial, as the mid-day peak and nighttime lull of

solar energy combined with peak electricity demand in the evenings causes the duck curve, which is known to cause utility challenges[41]. The daily dispatch profile on a peak-demand day in 2050 reveals that increased deployment of wave energy and (especially) offshore wind energy reduces the solar energy and storage charging peaks in the grid. As offshore wind and wave energy cost targets decline, we observe a maximum decrease of 26% in the solar generation peak on a peak-demand day in 2050. Contrary to solar energy, offshore wind, and wave energy are dispatched at an almost consistent level throughout the day, only decreasing when solar is in excess. Hence, the more consistent generation profiles of wave energy and offshore wind may be useful for serving the grid's base load and reducing the duck curve effect in a highly renewable grid.

As expected, lower cost targets of offshore wind energy result in more offshore wind installed capacity, as seen in Fig. 3a. We observe a maximum increase in installed capacity of offshore wind from 2 GW to 59 GW, which corresponds to an increase from 0.3% to 9% of the total installed capacity. Similarly, the amount of wave energy capacity installed in the Western Interconnection in each scenario increases as the cost targets of wave energy decrease, as seen in Fig. 3b. We observe a maximum increase in the installed capacity of wave energy from

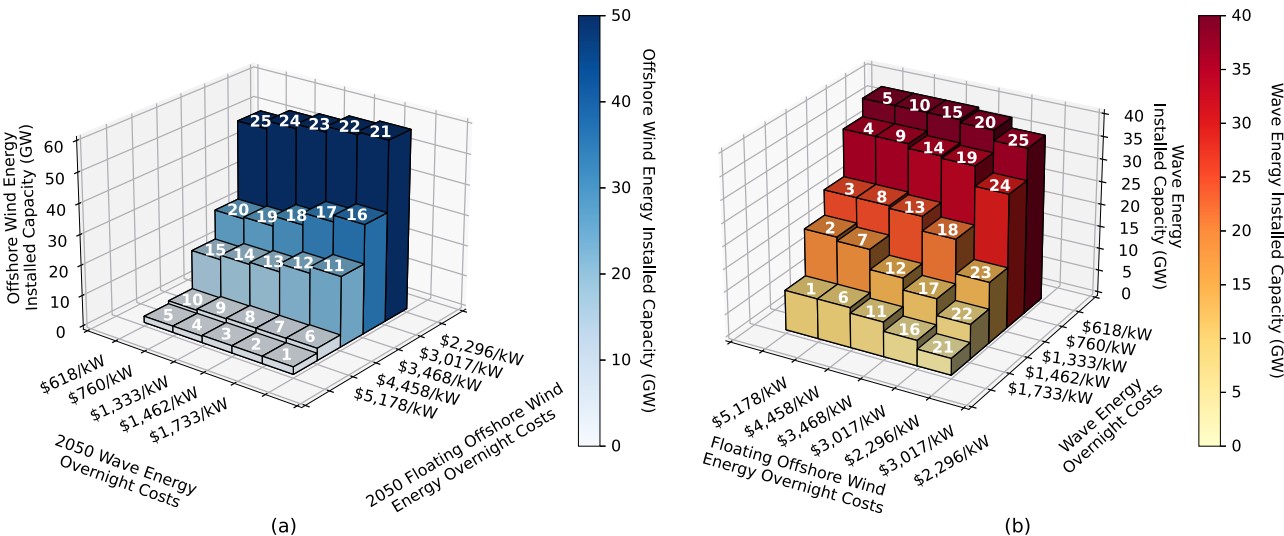

**Fig. 3 | 2050 offshore wind and wave energy capacity installed.** 2050 offshore wind energy (**a**) and wave energy (**b**) capacity installed (GW) in each scenario. Scenario numbers are displayed at the top of each bar. Offshore wind energy installed capacity increases as offshore wind energy cost targets decline and as wave energy cost targets rise. Wave energy installed capacity increases as wave energy cost targets decline and as offshore wind energy cost targets rise. *Note that the x and y axes are flipped in the plot on the left. This is done so that the trend is fully visible. Source data are provided as a Source Data file.

3.7 GW to 40 GW, which corresponds to an increase from 0.6% to 5.5% of the total installed capacity. The 40 GW of wave energy installed corresponds to 93% of the maximum amount of wave energy that could be installed in the 101 new wave energy candidate projects added to SWITCH WECC for this study. Hence, at the wave energy overnight and O&M cost targets of $618/kW and $13.25/kW, respectively, which corresponds to parity with utility PV in 2050, SWITCH nearly maxes out the amount of available wave energy capacity (as allotted by the candidate projects in study) on the U.S. West Coast.

Refer to Tables 3–10 of the Supplementary Information for detailed numerical results related to 2050 total installed capacity and installed capacity of individual technologies across the scenarios. Refer to Fig. 11 of the Supplementary Information for a graph of the 2050 peak-demand day dispatch profiles for the four edge-case scenarios.

**Highly intermittent sources in coastal zones are needed less**

Because offshore wind and wave energy farms deliver generated power to substations located along the U.S. West Coast, the generation profiles of coastal load zones (20 zones) are most affected by the deployment of these technologies. These impacts are significant to the whole system because the total electricity demand across the coastal load zones makes up almost 42% of the entire demand in the Western Interconnection in 2050. Figure 4 shows the daily dispatch profile that represents the dispatch specifically in coastal load zones on the peak day in December, 2050 in the four edge-case scenarios (with either highest or lowest cost target for offshore wind and/or wave energy). Similar to the total grid daily dispatch profiles, the daily dispatch profiles in the coastal load zones reveal that offshore wind and wave energy have almost constant generation throughout the day. This further suggests that they are well-suited technologies for serving the base and that they may contribute to less reliance on storage since less power is generated excessively during times of the day when it is not needed. Note that the load is the same in all scenarios, but that scenario 25 (least expensive offshore wind and wave energy cost targets) shows a notably smaller solar peak than scenario 1 (most expensive offshore wind and wave energy cost targets) (Fig. 4). Coastal load zones exhibit a maximum decrease of 24% in the solar peak on a peak-demand day in 2050. This indicates that the relatively constant nature of offshore wind and wave energy generation reduces the amount of

generation needed from more intermittent sources. For reference, the scenario 25 overnight cost targets of fixed-bottom and floating offshore wind and wave energy in 2050 are $1382/kW, $2296/kW, and $618/kW, respectively, while solar energy's 2050 overnight cost is assumed to be $703/kW. Notice that although the overnight cost of solar energy is significantly lower than that of offshore wind, we still observe a decline in the solar energy generation peak when offshore wind is most deployed. Coastal load zones in scenario 25 also have visibly fewer daily imports from other load zones and more daily exports to other load zones than in scenario 1, which implies that these zones are becoming less reliant on other zones to serve their local loads as more offshore wind and wave energy are deployed.

Figure 5 shows the monthly dispatch profiles in 2050 for coastal load zones for the edge-case scenarios. Between scenario 1 and scenario 25 (most and least expensive offshore wind and wave energy cost targets, respectively), there is a 31% decrease in annual (2050) energy imports (from other load zones) and a 58% increase in annual energy exports (to other load zones) in coastal load zones. We also observe this trend when only offshore wind or wave energy become dramatically cheaper while the other technology remains expensive.

These results reveal that wave energy and offshore wind deployment influence increases in energy exports from coastal load zones to other load zones and decreases in energy imports from other load zones to coastal load zones. Hence, if these technologies are deployed substantially, they may play a role in helping coastal regions to become more self-sufficient and also become larger generation centers for supporting inland regions.

We can observe the increase in offshore wind and wave energy generation along the U.S. West coast with decreasing cost targets when comparing the dispatch portfolio map of scenario 1 to the dispatch portfolio map of scenario 25 (Fig. 6). The decline in solar energy and energy storage dispatch along the coast is also visible.

California, which has the highest load, generation, and offshore wind and wave energy installed capacity when compared to the other coastal states, has an increased ratio of generation to load as offshore wind and wave energy are increasingly deployed. We observe the ratio of generation to load in California increase from 0.87 to 0.94. Washington (ratio increases from 0.78 to 0.84) and Oregon (ratio increases from 1.04 to 1.63) exhibit a similar pattern. This provides further evidence that when coastal states integrate more offshore wind

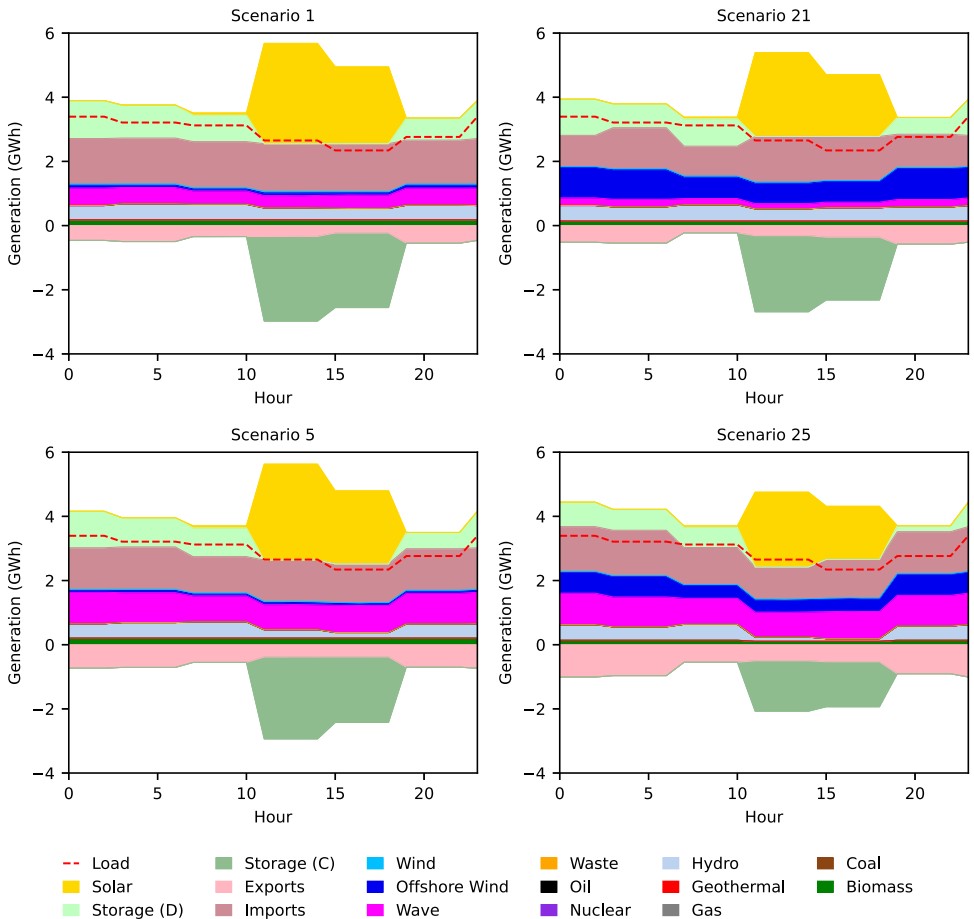

**Fig. 4 | 2050 daily (peak day Dec.) dispatch profiles in coastal load zones for four edge-case scenarios.** The daily dispatch profiles show relatively constant offshore wind (blue) and wave power (magenta) generation, decreased dispatch of solar energy (yellow) and energy storage (light green) with increased dispatch of offshore wind (blue) and wave energy (magenta), and decreased imports from other load zones (dark pink) and increased exports to other load zones (light pink) with increased dispatch of offshore wind and wave energy. Source data are provided as a Source Data file.

and wave energy into their electricity generation mixes, they are able to meet more of their state's demand and thus are more self-sufficient (i.e., less reliant on energy imports from other states).

Refer to Table 11 of the Supplementary Information to see the ratio of generation to load in California in 2050 overall 25 scenarios.

### Renewable energy curtailment increases

The constant nature of offshore wind and wave energy generation observed in Fig. 4 is an advantage that these technologies have over their renewable counterparts. Consequently, as offshore wind and wave energy are increasingly deployed, and hence more of the demand in coastal load zones is being met locally, there is more curtailment of land-based wind energy and solar energy in the year 2050. Figure 7 shows that as offshore wind and wave energy cost targets decrease, total curtailment in the grid increases by a maximum of 49 TWh (48% increase). This increase in curtailment corresponds to an increase of 3.4% in terms of percent of total available renewable electricity that is curtailed. More than half of that increase in renewable curtailment is attributed to increased curtailment of land-based wind energy and solar energy.

Refer to Table 12 of the Supplementary Information for detailed numerical results related to curtailment in 2050.

### More offshore wind energy leads to more built transmission

We observe that lower wave energy cost targets lead to less installed land-based transmission in 2050, and lower offshore wind energy costs lead to more installed land-based transmission (Fig. 2d). Note that the

cost of underwater transmission for the offshore wind and wave energy deployments is captured in the connection cost of each project, but the analysis in this section focuses on installed capacity of land-based transmission only. All subsequent discussions of transmission are referring to land-based transmission. Across the scenarios where wave energy cost targets are decreasing, the amount of installed transmission decreases by a maximum of 21.8 GW (15% decrease). Across the scenarios where offshore wind energy cost targets are decreasing, the amount of installed transmission increases by a maximum of 80 GW (92% increase). The increase in transmission associated with increasing amounts of offshore wind energy installed can be explained by the new transmission required to transport power produced in offshore wind farms throughout the main grid. If significant amounts of offshore wind generation cause the coastal load zones to become larger generation centers, then increased transmission capacity will be needed to move power from the coasts inland, as observed in Fig. 6.

The decrease in installed transmission with lower wave energy costs becomes more prevalent once offshore wind energy cost targets reach the moderate NREL ATB scenario or lower targets. This trend is explained by the decreased offshore wind energy capacity as wave energy cost targets decrease. This relationship between installed capacity of offshore wind energy and installed transmission capacity can be observed by comparing Figs. 3a and 2d. This suggests that offshore wind energy is the main diver for increased installed transmission, and that installing a more even mix of offshore wind and wave energy, rather than installing a significant amount offshore wind

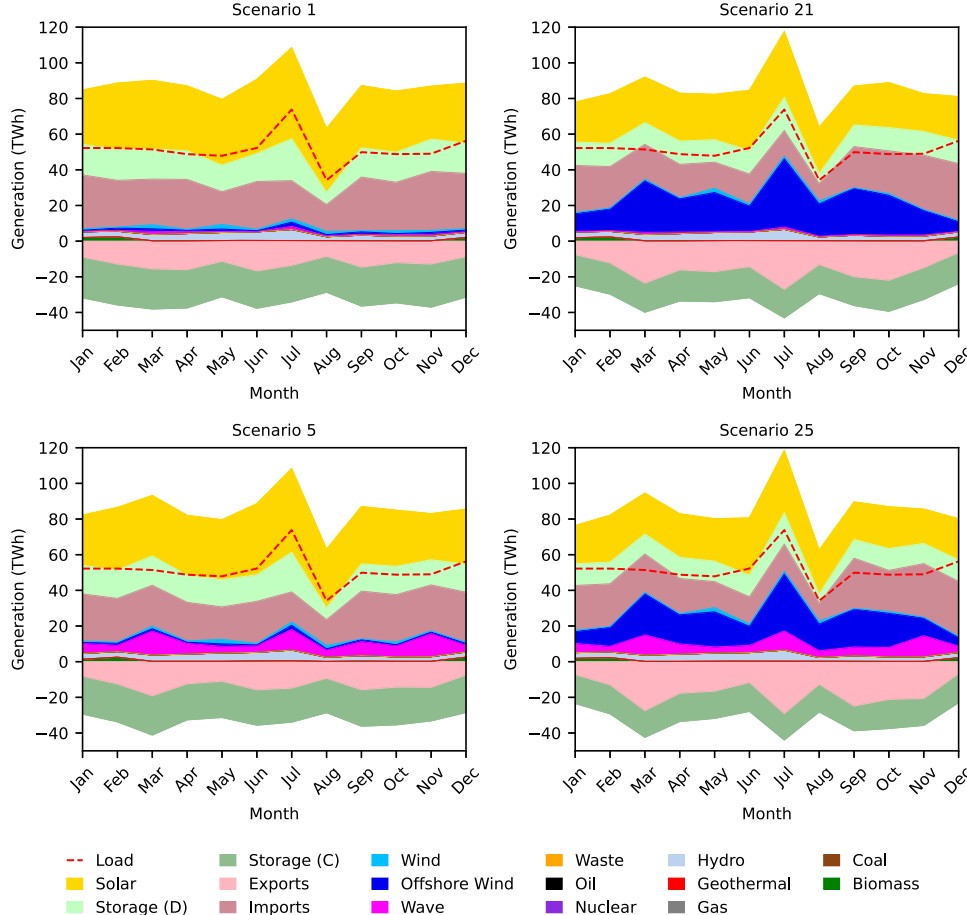

**Fig. 5 | 2050 monthly dispatch profiles in coastal load zones for four edge-case scenarios.** The monthly dispatch profiles show decreased dispatch of solar energy (yellow) and energy storage (light green) with increased dispatch of offshore wind (blue) and wave energy (magenta), as well as decreased imports from other load zones and increased exports to other load zones (shades of pink) with increased dispatch of offshore wind (blue) and wave energy (magenta). Source data are provided as a Source Data file.

energy and a small amount of wave energy, can help decrease the amount of new transmission required.

Refer to Table 13 of the Supplementary Information for detailed numerical results related to 2050 installed transmission capacity across the scenarios.

**Lower cost targets lead to more collocation**

Finally, we observe that lower offshore wind and wave energy costs lead to more collocated offshore wind and wave energy farms (Fig. 8). Scenario 1 (most expensive offshore wind and wave energy cost targets) has no collocated offshore wind and wave energy farms in 2050. However, as offshore wind and wave energy costs decline, we see the number of collocated sites increase to a maximum of 28 (out of 101 total possible sites for collocation), which corresponds to 23% of the installed offshore wind and/or wave energy farms exhibiting collocation.

This increase in the number (and percent) of collocated sites as offshore wind and wave energy become increasingly more cost competitive with other renewable resources shows that the optimization model favors the collocation of these technologies as they are increasingly deployed for electricity generation. Although, not captured in our model, this tendency would be even stronger as we would expect to observe reduced costs associated with collocated offshore wind and wave energy as they would share land-based infrastructure.

Refer to Tables 14–17 of the Supplementary Information for detailed numerical results related to collocation of offshore wind and wave energy across the scenarios.

**Total system cost decreases by up to 4%**

We observe that as offshore wind and wave energy become increasingly cost competitive with other renewable technologies, and consequently become increasingly deployed in the grid, the total Western Interconnection System cost in net present value (NPV), with 2018 as the dollar base year and summed across all four investment periods, decreases by a maximum of 4%.

There are several factors that contribute to this decline in addition to decreasing offshore wind and wave energy costs targets. Firstly, 2050 incurred fuel costs decline slightly across the scenarios (maximum decrease of 0.9%). The main contributing factor to this decline in fuel costs is a 7.8% decrease in biomass generation in the year 2050. Second, incurred energy storage fixed costs decline by a maximum of 50%. This is a direct result of the significant decrease in installed energy storage that is observed with increased penetration of offshore wind and wave energy. Third, incurred O&M and fixed costs of electricity generators slightly decline (maximum decrease of 1.4% and 2.3%, respectively). This is likely a consequence of the over 17% reduction of installed capacity in the grid, as well as the lower investment and O&M costs assumed for offshore wind and wave energy. In contrast, we observe a maximum increase of 28% in transmission fixed costs, which is a direct result of the increase in transmission capacity observed with decreasing offshore wind costs. However, decreased wave energy cost targets consistently cause decreased transmission costs. This is due to the reduced transmission required when more wave energy is deployed, as explained in the section titled, "More Offshore Wind Energy Leads to More Built Transmission."

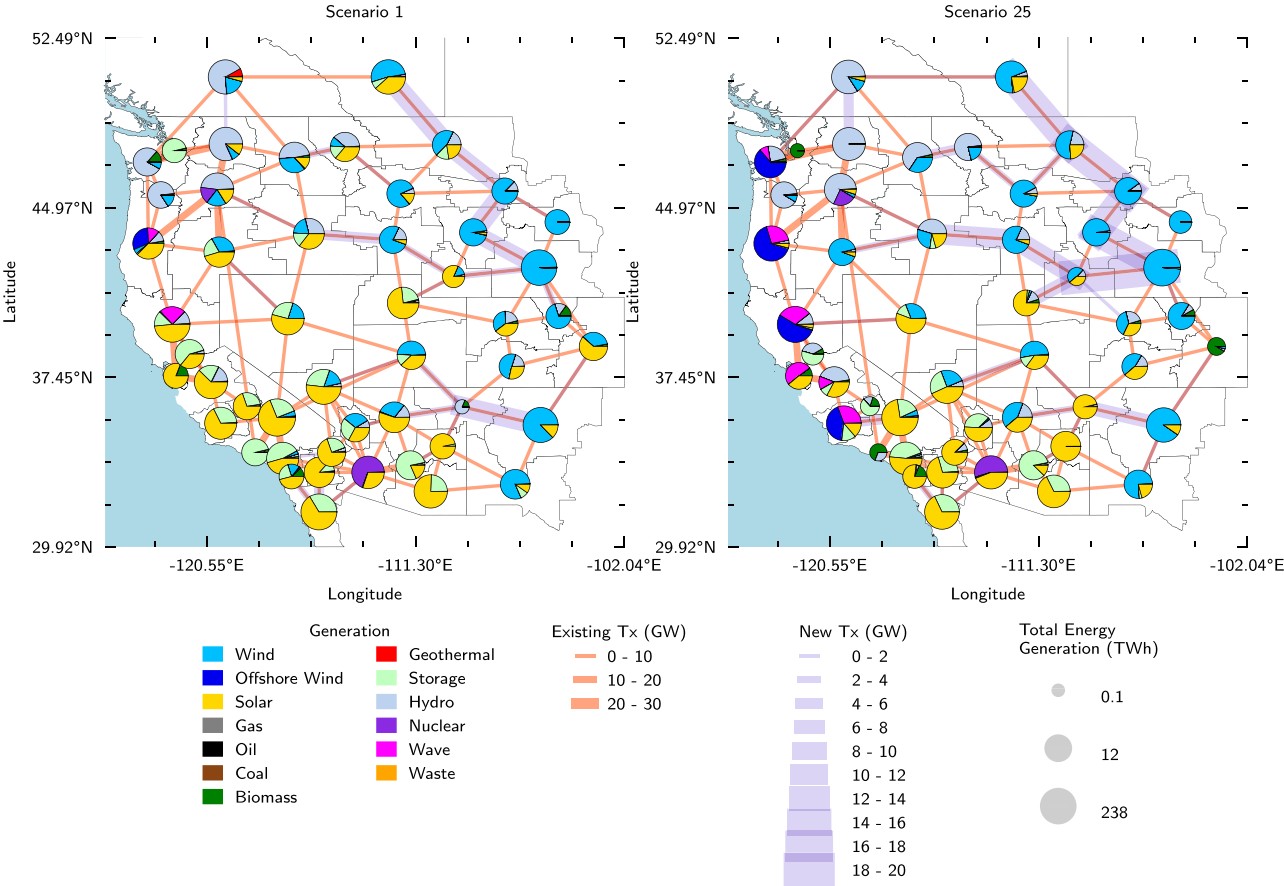

**Fig. 6 | Annual generation and transmission lines deployment.** Scenario 1 (left) and scenario 25 (right) (most and least expensive offshore wind and wave energy cost targets, respectively) annual generation breakdown and transmission lines for each load zone in 2050. Between scenarios 1 and 25, coastal load zones show an increase in the share of electricity generation from offshore wind and wave energy, as well as less generation from solar energy and energy storage. Source data are provided as a Source Data file.

Refer to Table 18 of the Supplementary Information for detailed numerical results related to system cost components across the scenarios.

## Discussion

As we have seen, even relatively small percentages of offshore wind and wave energy penetrations in a 2050 zero-emissions electricity mix have significant implications on the grid. One of the most remarkable consequences of deploying offshore wind and wave energy we observe in this study is the large (133 GW, or 17.3%) reduction between the scenarios in total installed generation capacity in a 2050 zero-emissions Western Interconnection. This decrease in installed capacity is tied to less installed capacity of renewable resources with intermittent diurnal generation patterns, such as solar energy, and consequently less energy storage. This implies that these technologies will play a key role in limiting the upsizing of generation capacity in the grid, therefore limiting costs, as we move away from fossil fuels. The results show that for offshore wind and wave energy to induce >10% reduction in the 2050 total system installed capacity, offshore wind energy costs would have to decline to those of the NREL 2022 ATB advanced scenario, and wave energy costs would need to decline by at least 50%. The U.S. Office of Energy Efficiency and Renewable Energy has even more ambitious cost targets for floating offshore wind turbines over the next decade: The Floating Offshore Wind Energy Shot Initiative seeks to lower LCOE costs of offshore wind turbines by more than 70%, to $45/MWh by 2035[42]. This is $7/MWh less than what the NREL 2022 ATB advanced scenario assumes floating offshore wind will cost by 2035. According to NREL, for the ATB advanced scenario to

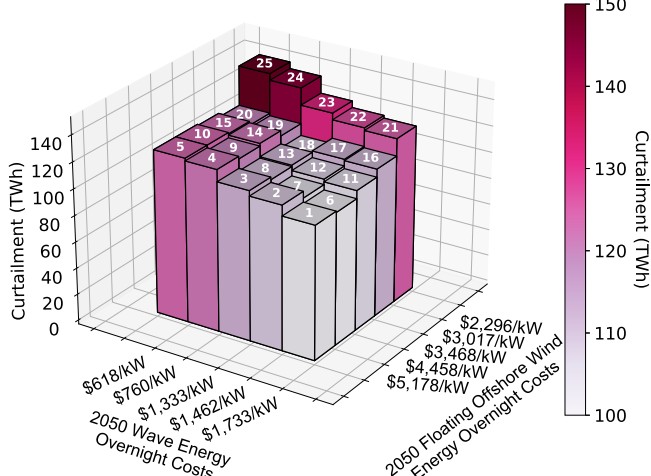

**Fig. 7 | 2050 energy curtailment (TWh) in the Western Interconnection for each scenario.** Energy curtailment increases with increasing offshore wind and wave energy cost targets. Scenario numbers are displayed at the top of each bar. *Note that the x axis and y axis are flipped with respect to plots **a** to c in Fig. 2. This is done so that the trend is fully visible. Source data are provided as a Source Data file.

manifest, turbine sizes would need to increase at a rate that is considerably higher than in recent years[40]. Offshore wind energy innovation that leads to cost reductions also includes significant changes to the manufacturing, installation, operation, and performance of wind

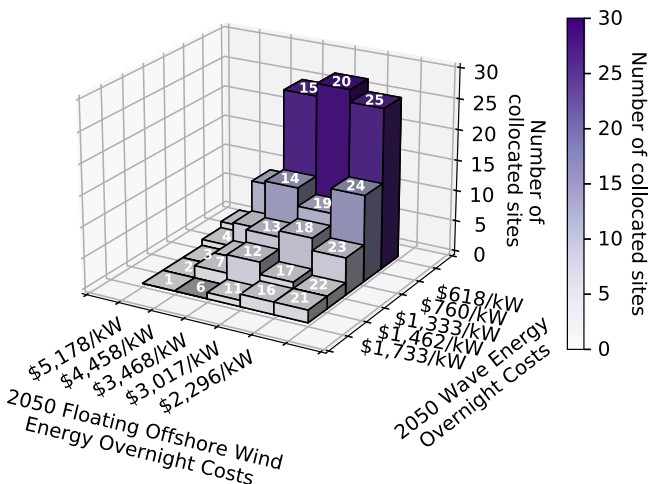

**Fig. 8 | Number of sites chosen for collocation in 2050 in each scenario.** In general, the number of sites chosen for collocation increases with decreasing offshore wind and wave energy cost targets. Scenario numbers are displayed at the top of each bar. Source data are provided as a Source Data file.

farms[40]. Wave energy would need to experience similar innovations to achieve these cost reduction targets, and the supply chain and manufacturing infrastructure to support the deployment of Wave Energy Converters (WECs) would need to be established. Fortunately, as offshore wind energy becomes increasingly deployed, it is likely that the infrastructure built to enable its adoption will positively influence wave energy cost reductions since the two technologies share the same ocean environment and require similar infrastructure.

While offshore wind and wave energy have faced many cost challenges associated with the hostility of ocean environments, their costs are expected to decline as supply chain infrastructure becomes more developed, designs become more efficient and refined, and markets adapt to compensate offshore wind and wave energy developers appropriately for the ancillary services that these technologies may provide to the grid (such as base-load support, reactive power support, etc.). Wave energy in particular has a long way to go when it comes to reducing costs by designing standardized WECs that can be manufactured with streamlined techniques, a problem further complicated by the fact the optimal WEC design can vary depending on the dominant wave frequency at a given site[23]. Standardization may still take place to create main categories of commonly used WEC designs which can be refined individually and deployed in areas with corresponding ocean conditions. Hence, there is still significant room for improvement of these technologies and the physical, economic, regulatory, and political infrastructure to support them.

Because the benefits that offshore wind and wave energy may provide for the future zero-emissions grid are so significant, we recommend that policy makers design incentives to stimulate investment, research, and development of these technologies that will drive their overnight and O&M costs sufficiently down to ensure that they become cost competitive with other renewable resources. For wave energy to reach cost parity with land-based wind by 2050, its overnight and O&M costs would need to decline by ~80% and 70%, respectively, over the next three decades. For offshore wind energy to align with the NREL 2022 ATB advanced scenario costs by 2050, fixed-bottom and floating offshore wind energy overnight and O&M costs would need to decline by ~43% over the next three decades. Although these declines seem drastic, solar and land-based wind energy have demonstrated momentous cost declines in the past decade. Solar energy costs have decreased 80% and land-based wind energy costs have decreased almost 40% since 2010[43]. Hence, significant drops in renewable energy costs are not unprecedented.

Additionally, the results of this study show that reduced offshore wind and wave energy costs result in increased collocation of the technologies in an optimal grid. Research suggests that through collocation, grid infrastructure, O&M, and licensing expenses could all be shared (which is not captured in this study). For instance[44], shows that two-thirds of offshore wind farm project development costs can be shared with wave energy deployments. However, the research on assessing collocation potential for offshore wind and wave energy in the Pacific Ocean is severely limited. Capacity expansion planners and offshore wind and wave energy developers should consider co-design and the potential for collocation of these technologies to avoid costly and inefficient integration in the future. Hence, if we are to utilize the synergistic benefits of offshore wind and wave energy in the most efficient way, there should be significant funding for research surrounding collocation of these technologies in the coming years.

## Methods

### Overview

First, we identify sites with high potential of offshore wind and wave energy along the coast of the Western Interconnection. We next model candidate generation projects at these sites, (i) filtering out sites that are in marine protected areas (MPAs) with strict classifications[45], military danger zones and restricted military activity areas[46], and (ii) calculating the hourly capacity factors for each candidate project for one year of data. Finally, we use SWITCH, a power system capacity expansion model, to study the role and impacts of these offshore wind and wave energy candidate projects under 25 scenarios with different cost targets.

A more detailed overview of the methodology for this study is summarized in Fig. 9.

### Data acquisition and processing

The sites of industry interest represent high-potential wave farm sites along the U.S. West Coast. They are calculated as the result of a scoring framework developed by CalWave. Each site considered by CalWave receives a score between 0 and 100, based on a weighted sum of the following six quantitative parameters: wave energy resource density, distance to shore, water depth, wind resource, bathymetry, and local population density[47]. The parameters are weighted based on CalWave's assessment of their relative importance to the development of utility-scale wave energy infrastructure. CalWave uses NREL's report on Marine Hydrokinetic Energy Site Identification and Ranking[48] as a guideline for their own ranking framework. The parameters that CalWave considers which coincide with NREL's report are wave resource density and water depth. Some differences between the parameters considered by CalWave and NREL are as follows:

- While NREL considers market size and distance to transmission connection, CalWave considers local population density and distance to shore
- NREL considers energy price and shipping cost, but CalWave does not
- CalWave considers bathymetry and wind resource, while NREL does not
- NREL assigns equal ranking to each of the parameters considered, while CalWave assigns different weights to each parameter empirically based on their experience as a wave energy developer and input from wave energy industry and academic ocean energy experts. The parameters in order of assigned weight from highest weight to lowest weight is as follows:
  1. Wave resource
  2. Distance to shore
  3. Water depth
  4. Wind resource
  5. Bathymetry
  6. Local population density

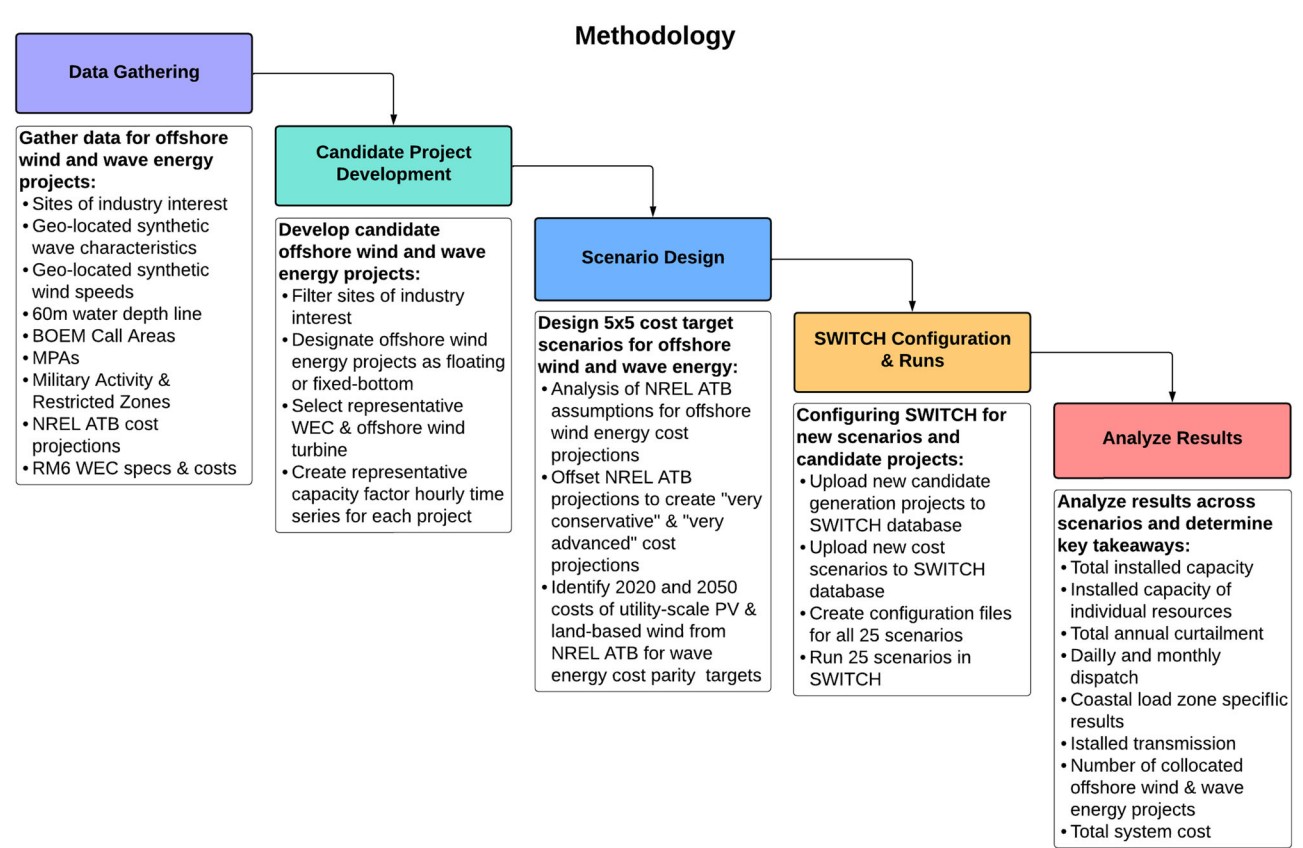

**Fig. 9 | Methodology.** Overview of methodology used for this study.

It is important to mention that the CalWave scoring framework does not use costs and existing infrastructure as compared to the report from NREL[48] because it intentionally encourages the development of wave energy infrastructure in the locations most technically suitable. The sites that rank as the top 100 sites according to CalWave's proprietary framework are identified as the industry sites of interest for the U.S. West Coast. Figure 10 shows each site of interest represented by the latitudinal and longitudinal coordinates of its center (blue points).

When developing candidate projects for offshore wind and wave energy, we first filter the sites of industry interest to ensure that no site overlaps with MPAs that have the 3 strictest classifications[45]: No Take, No Impact, No Access. No Take zones "prohibit the extraction or significant destruction of natural and cultural resources," No Impact zones "prohibit all activities that could harm the site's resources or disrupt the ecological and cultural services they provide," and No Access zones "restrict all human access in order to prevent potential ecological disturbance"[45]. Additionally, the sites are filtered to ensure that no site overlaps with military danger zones and restricted military activity areas. No United Nations Educational, Scientific and Cultural Organization (UNESCO) World Heritage Marine Sites (WHMSs) overlap with any of the sites of industry interest[49]. Four sites of industry interest overlap with these restricted zones, thus they are removed from consideration for candidate project locations.

Sites with an ocean depth 60 m or shallower are classified as fixed-bottom offshore wind resources, and sites with an ocean depth deeper than 60 m are classified as floating offshore wind resources[40]. It is important to make this distinction because fixed-bottom and floating offshore wind farms have different cost targets and technical characteristics. In order to give each site an area in which arrays of wind turbines and WECs can be installed, rectangular polygons are drawn around each site of industry interest using QGIS. Figure 10

shows these candidate project areas along the U.S. West Coast. Each polygon is designed such that no site areas overlap, no MPAs of restricted classification or military activity zones are encroached on, and each area falls exclusively in shallow (≤60-m depth) or deep (>60-m depth) water. The polygons are drawn such that their length is parallel to the coastline since waves tend to form parallel to the coastline. Some polygons in Fig. 10 are so small that they may not appear visible, but note that all industry sites of interest are given a corresponding candidate project area. Some sites that are very close to the coast have limited areas that they could encompass because of nearby land in the east direction and deep water in the west direction.

Five U.S. West Coast offshore wind Call Areas[50,51] (Coos Bay, Brookings, Humboldt, Morro Bay, and Diablo Canyon) are added to the list of candidate projects, bringing the total number of candidate project areas to 101. Call Areas are potential commercial offshore wind development areas identified by the Bureau of Ocean Energy Management (BOEM) for public comment during the Call for Information and Nomination stage[50]. The offshore wind Call Areas are important to include as candidate project areas for this study so that the potential for offshore wind, wave energy, and collocated offshore wind and wave energy may also be evaluated for these federally identified sites from a grid capacity expansion planning perspective, in addition to the wave energy sites of industry interest. The largest candidate project area (pink polygons in Fig. 10) is designed to be no larger than the largest offshore wind Call Area.

We do not enforce a maximum water depth on the offshore wind and wave energy candidate projects because it is uncertain what water depths will be possible to install marine energy devices in the year 2050 due to technological advancements over the coming decades. Furthermore, the BOEM Offshore Wind Call Areas are between 200 m and 1300 m deep. Less than 5% of the candidate project areas have any portions of their areas beyond the 1300 m depth contour.

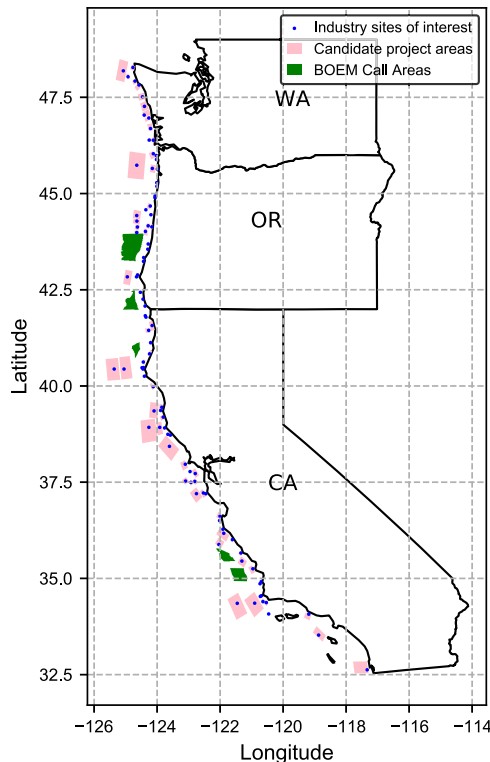

**Fig. 10 | Sites of industry interest (blue points) along the U.S. West Coast, offshore wind and wave energy candidate project areas (pink polygons), and BOEM Call Areas (green polygons).** The sites of industry interest appearing in this figure have been filtered to exclude sites in MPAs and military danger zones. All candidate project areas and BOEM call areas may have offshore wind energy, wave energy, or both technologies installed. Source data are provided as a Source Data file.

Refer to section 1 of the Supplementary Information for more details related to the methodology for candidate project design. There we include the names and coordinates of the sites removed from consideration (Supplementary Table 1) and details regarding what characteristics were considered during the sites of interest filtering process.

Wave energy availability can be measured using the significant wave height ($H_s$) and energy period ($T_e$) of a wave. These metrics serve as input data for determining how much power a WEC can generate. We use all 699,903 coordinates available along the U.S. West Coast from the U.S. Department of Energy (DOE) Water Power Technology Office's (WPTO) U.S. Wave dataset[52]. This dataset is the highest spatial resolution publicly available long-term (1979-2010) wave hindcast dataset[52]. It has an unstructured grid spatial resolution that ranges from 200 meters (in shallow water) to 10 kilometers (in deep water)[52]. The 699,903 available data points are generated from the SWAN and WaveWatch III models, which have been validated using publicly available spectral data from buoys[53].

We overlay these coordinates with the candidate project areas (Fig. 10) in QGIS to identify 89,650 overlapping coordinates. We use 3-hour time resolution time series of wave characteristics for the year 2006 corresponding to every ten (to reduce download time) of the 89,650 coordinates from[52]. A total of 8811 coordinates are downloaded, and each coordinate has a time series that includes timestamps, significant wave height values in meters, energy period values in seconds, and latitude/longitude coordinates associated with the locations for which data is extracted. We linearly interpolate to convert the time resolution of the dataset from 3-hour to 1-hour resolution. Due to the high spatial resolution of the 3-hour dataset, the linearly

interpolated data is used to develop the wave characteristic time series used in this study. We assign a time series to each wave energy candidate project by taking the average time series of the WPTO coordinates within each project area.

The capacity factor, $CF$, is defined as the ratio between the available generating power, $P_g$, and the rated power capacity, $P_r$, as shown in Eq. (1).

$$CF = \frac{P_g}{P_r} \qquad (1)$$

Since the capacity factor of a WEC is subject to the availability of the primary resource (e.g., wave energy), the capacity factor changes according to the wave characteristics at the location where the WEC is installed at a given time.

In this study, we choose the Reference Model 6 (RM6) Oscillating WEC as the representative WEC[54]. Its rated power capacity is 350.5 kW and its power matrix can be downloaded from NREL's Marine Energy Atlas[55]. The power matrix reveals the available generating power of the WEC as a function of the significant wave height (meters) and the energy period (seconds).

We use the wave height and energy period data from the linearly interpolated 1-hour time resolution time series, the RM6 power matrix, and Eq. (1) to calculate hourly capacity factors corresponding to the WPTO coordinates. We calculate an average hourly time series for the year 2006 corresponding to each candidate project area by averaging the time series of all of the WPTO coordinates that fall within each area. We do not consider the wake effects of WECs because there is limited information on this topic, and wake effects can vary largely from one WEC design to another.

In order to determine the maximum possible installed wave energy capacity at each site, we assume the packing density of the WECs to be 1.0515 MW/km². To derive this value, we consider the array layout design provided by the RM6 report[54]. We calculate the packing density as follows (2):

$$\frac{3\,\text{WECs}}{1\,\text{km}^2} \times \frac{350.5\,\text{kW}}{1\,\text{WEC}} \times \frac{1\,\text{MW}}{1000\,\text{kW}} = 1.0515\,\frac{\text{MW}}{\text{km}^2} \qquad (2)$$

Refer to section 1 of the Supplementary Information to see the details of two error analyses related to the wave energy capacity factor time series used in this study:

1. To justify taking every 10 of the overlapping points
2. To verify that linear interpolation from 3-hour to 1-hour resolution for the wave energy capacity factor time series does not introduce substantial error

We choose the 2020 ATB Reference 15 Wind Turbine and its corresponding power curve as the representative wind turbine and power curve for this study[56]. This is the same turbine used by NREL to develop the moderate-cost target for offshore wind in the 2022 ATB[40]. It has a rated power of 15 MW, a height of 150 meters, and a rotor diameter of 240 meters[56]. Similarly to the wave energy data, we extract the coordinates of the NREL Offshore NW Pacific Dataset[57] for 160-meter height, and we overlay the coordinates with the candidate project areas shown in Fig. 10 to determine which coordinates overlap. We download hourly time series data of wind characteristics for all coordinates that lie within the areas. The NREL Offshore NW Pacific Dataset is a 21-year wind resource dataset with a 5-minute time resolution created using the Weather Research and Forecasting numerical weather prediction model[57].

We design 101 offshore wind candidate projects to occupy the same areas as the wave energy candidate projects to allow the potential for collocation of these technologies. We create an interpolation function using the power curve of the turbine while

considering the turbine's operating limits to determine the power generated by the turbine at any given wind speed in m/s. We assign a time series to each offshore wind candidate project by taking the average time series of the NREL coordinates within each project area. A total of 9207 coordinates lie within the project areas. We separate them based on which project area they fall within and use them to calculate an average time series for each area. We compute the hourly offshore wind energy capacity factors as the ratio between the available generating power and the rated power capacity of the turbine (Eq. (1)).

In order to determine the maximum possible installed offshore wind energy capacity at each site, we assume the packing density of the offshore wind turbines to be 4.3 MW/km². This value is based on the average theoretical capacity density of the Morro Bay Wind Energy Area[58], which is a current offshore wind leasing area on the U.S. West Coast. There is no standard for offshore wind turbine spacing because packing density can vary based on site-specific conditions or farm designs. Thus, for simplicity, we assume the same packing density for all fixed-bottom and floating turbines. Furthermore, we do not consider wake effects of offshore wind turbines given that this is a variable dependant on specific farm array design that can be minimized by developers through strategic design.

### SWITCH model

SWITCH[36] is a linear programming electricity capacity expansion model that finds the least-cost generation portfolio and transmission infrastructure subject to electricity demand and operational constraints. SWITCH is able to model multiple investment periods (periods of one or more years where investment decisions are made), e.g., sets of decades, and multiple time series (chronological sequences of grouped timepoints where operational decisions are made) with different time resolution for each investment period.

The objective function minimized corresponds to the total power system cost, i.e., investment and operational costs of generation and transmission. The decision variables of the optimization problem can be summarized in the following sets: capacity investment decisions for each potential new generation project in each period, capacity investment decisions for each potential new or existing transmission line between any load areas in each period, hourly dispatch decisions for each existing and new generator installed for each period, and decisions on hourly transmitted energy through the existing and new transmission lines.

The main constraints in the optimization problem are: power balance in each zone where power generators, storage technologies, demand and transmission lines are connected, electricity dispatch of the generation technologies limited by their corresponding power capacities, energy flows across the transmission lines limited by their corresponding power capacities, electricity dispatch of renewable energy generators also limited by geolocated hourly capacity factor time series, generation from each hydropower plant limited by historical monthly availability (minimum, average and maximum generation), biomass and geothermal deployment limited by the resource availability, respect yearly maintenance time for each generation technology, policy constraints as carbon cap, carbon tax, Renewable Portfolio Standards, among others. For its detailed mathematical description, refer to section 6 of the Supplementary Information.

Many research groups have further developed different versions of the SWITCH model to analyze decarbonization pathways in different regions[1,37–39,59–64]. We use the SWITCH WECC[65] model which represents the Western Interconnection by dividing it into 50 geographical zones. The time resolution can vary from hourly to sampled hours that represent typical days during the years being optimized. These modeling virtues of the SWITCH WECC model allow a more realistic study of the expansion and operation of large regional electrical grids with the presence of renewable intermittent resources.

As mentioned previously, investment decisions are made in periods 2020, 2030, 2040, and 2050 which result in a zero-carbon grid by 2050. Our analysis in the Results section focuses on results in 2050. As a reminder, we represent each period as ten-year periods by sampling every month in 2020, 2030, 2040, and 2050, two days per month (median and peak load days) and every four hours per day (12 months × 2 days/month × 6 hour/day = 144 hours). Peak days have a weight of one and median days of $n−1$ where $n$ is the number of days of that month, and this represents a full month.

The use of a four-hour interval instead of the typical hourly dispatch is part of the reason high geographic resolution could be achieved. Additionally, the reduced complexity from using a four-hour time interval allows us to spend more computational effort on having a high geographical resolution for potential sites and having 2030, 2040, and 2050 investment decisions to better understand the transition. A faster run time from sampling hours also allowed us to create many scenarios to evaluate the relative deployment of offshore wind and wave energy.

We model the transmission system of the Western Interconnection using Ventyx geolocated aggregated transmission line data[66] and the thermal limits from the Federal Energy Regulatory Commission[67]. In total, we consider 105 existing transmission lines connecting load zones of the Western Interconnection. SWITCH can decide to build more transmission lines or expand the capacity of existing ones if it is optimal. The model considers transmission line derating and losses.

The electricity demand profiles come from historical hourly loads from 2006[68,69] (and ITRON consulting group). These profiles are projected for future years. The model includes geolocated hourly capacity factor time series for over 7000 potential new locations for solar and land-based wind power, as well as potential new locations for other renewable energy technologies (geothermal and biomass). New power plants for nuclear energy, hydropower, and geothermal energy are also included as candidate projects, as well as battery energy storage and pumped hydro storage. We calculate hourly existing and potential new land-based wind farm power output from the 3TIER Western Wind and Solar Integration Study wind speed dataset[70,71] using idealized turbine power output curves on interpolated wind speed values. For existing and potential new solar power plants, we simulate the hourly capacity factors of each project over the course of the year 2006 using the System Advisor Model from NREL[72]. The optimization can then choose from over 7000 potential new geolocated generators in the Western Interconnection. Fuel price projections for each load area are from the U.S. Energy Information Administration[73]. Capital costs and O&M costs are from NREL ATB 2020[74]. The historical pool of exiting power plants in the Western Interconnection is from the U.S. Energy Information Administration (EIA-860, EIA-923, 2020 release[75]).

### Scenarios description

We seek to evaluate the role that offshore wind and wave energy may play in decarbonizing the Western Interconnection by the year 2050. Because the objective function in SWITCH minimizes system cost, we expect deployment of offshore wind and wave energy to vary with cost. Therefore, we design twenty-five scenarios with different offshore wind and wave energy 2050 cost targets. All costs are reported in 2018 U.S. dollars (USD), which is the base year we use in SWITCH WECC. The 2020 wave energy overnight and O&M costs for all scenarios are $3465/kW and $105.4/kW, respectively. We compute these values by dividing the estimated RM6 WEC overnight and O&M costs for 10-unit deployment by 10[54]. This division by 10 is justified by the economies of scale of the candidate projects designed for this study: the reported costs assume a 10-unit deployment while the designed wave energy candidate projects may have several hundreds of WECs deployed in each site area (based on the packing density assumed and

the size of the candidate project areas). The RM6 report[54] demonstrates how lower costs are associated with larger-scale WEC farms. These assumed 2020 wave energy costs align with the lower-end of a range provided by leading wave energy developers as an approximation of the current capital expenditure and operating expenditure costs of wave energy[76].

As a reminder, there are five different cost targets for wave energy, with the most conservative cost target corresponding to a 50% cost reduction by the year 2050 and the most optimistic cost target corresponding to parity between the 2050 overnight and O&M wave energy costs and the NREL 2022 ATB[40] utility-scale PV energy projected 2050 costs (Fig. 1). As mentioned previously, we assume a linear projection between the wave energy 2020 and 2050 costs. Although we could use learning coefficients to model the decline in wave energy costs between 2020 and 2050, formulating accurate cost projections (or learning/experience curves) is not within the scope of this work, but it may be considered in future work. Additionally, the study in[77] uses a two-stage Monte Carlo simulation to forecast the levelized cost of electricity (LCOE) for wave energy and finds that the cost reductions are nearly linear. Thus, we assume a linear trend for its simplicity.

Similarly, we design five offshore wind overnight and O&M cost targets based on the NREL 2022 ATB cost projections for fixed and floating offshore wind energy (Fig. 1). We use Wind Resource Class 3 for fixed-bottom turbines, and Wind Resource Class 12 for fixed-bottom turbines. According to NREL, Class 3 and Class 12 are the most representative of near-term U.S. fixed-bottom and mid-term U.S. floating offshore wind projects, respectively[40]. We design an additional very conservative offshore wind cost target such that $488.39/kW is added to the overnight costs and $15.90/kW-yr is added to the O&M costs of the NREL 2022 ATB conservative projection for fixed-bottom offshore wind turbines (after converting to 2018 dollars). For floating offshore wind turbines, we add $720.55/kW to the overnight costs and $14.98/kW-yr to the O&M costs (after converting to 2018 dollars) to generate a very conservative scenario. Similarly, we design an additional very advanced offshore wind cost target such that $488.39/kW is subtracted from the overnight costs and $15.90/kW-yr is subtracted from the O&M costs of the NREL 2022 ATB advanced fixed offshore wind projection (after converting to 2018 dollars). For floating offshore wind turbines, we subtract $720.55/kW from the overnight costs and $14.98/kW-yr from the O&M costs of the NREL 2022 ATB advanced floating offshore wind projection (after converting to 2018 dollars). The offsets of $488.39/kW, $15.90/kW-yr, $720.55/kW, and $14.98/kW-yr are chosen because they equal the average difference between the NREL 2022 ATB moderate and advanced scenario overnight and O&M costs and the moderate and conservative scenario overnight and O&M costs for fixed and floating offshore wind turbines, respectively.

As mentioned previously, the five wave energy cost targets and five offshore wind cost targets are combined into 25 (5 × 5) scenarios, as shown in Fig. 1. Refer to Figs. 4–5 of the Supplementary Information for alternate visualizations of the cost target scenarios.

All offshore wind and wave energy candidate projects assume an interconnection cost of $487,000/MW of capacity installed. This is the average interconnection cost for an offshore wind project with a commercial operation date of 2023 in 2018 dollars from ref. 78.

## Limitations

Since the optimization model chooses to collocate more offshore wind and wave energy projects as their costs decrease, we infer that pairing the technologies could be valuable in a zero-emissions grid due to the shared land-based infrastructure cost savings that are achieved when the technologies are collocated. One limitation of this study is that it does not capture the cost benefits associated with shared underwater transmission infrastructure in collocated offshore wind and wave energy farms. Future work is planned to additionally capture this benefit of collocation in the model, as well as to distinguish connection costs across offshore energy sites according to each site's bathymetry.

Another limitation is that existing underwater pipelines and cables, existing shipping routes, and archeological sites other than those included in UNESCO's WHMSs are not considered in the process of filtering sites of industry interest. We also do not consider the proximity of each site to residential areas on land. We believe that the potential impact from not including these parameters when filtering the industry sites of interested is limited but of local relevance. The main difference in the study if these parameters were able to be included would likely be the shapes of the individual candidate project areas (if they are modified to avoid additional ocean zones). Although this may slightly alter the capacity factor time series of certain candidate project areas, we believe it is unlikely that it would substantially change the system-wide trends we observe when integrating various amounts of wave and offshore wind energy into the Western Interconnection. One aspect that could have a larger impact is if more areas are classified as not suitable for economic activity due to ecological considerations. In that case, our study can show how to prioritize deployment if less offshore energy installed capacity can be deployed.

Furthermore, our study does not enforce a maximum water depth on the sites of industry interest, because it is unclear what the limit of water depth for floating offshore wind turbines will be in the year 2050. Although only 5 of the 101 candidate project areas have any portions of their areas beyond 1500 meters of depth, these sites may be challenging to develop due to their extreme bathymetric conditions and significant distances from shore. Hence, the inclusion of these 5 sites without adding a cost multiplier to account for their innate deployment challenges slightly diminishes the realisticness of the model. However, since less than 5% of the sites exhibit very deep water, we believe the impact is minimal.

Lastly, the temporal resolution of our study (2 representative days per month, every 4-hours) may not fully capture the unique power output qualities of offshore wind and/or wave energy in the model. Although the simplified temporal resolution allows us to run a larger set of scenarios with very high spatial resolution, it diminishes the amount of information that the model draws from the capacity factor time series for each marine generator. We believe that a higher temporal resolution would lead to the same trends observed in this study, with slight variations in the numerical results. In our previous study that uses SWITCH to evaluate the impact of using various time sampling resolutions on the utilization of long-duration storage (LDS)[79], we find that although the utilization of LDS is affected by the time sampling resolution used, the overall installed capacity mix does not vary largely between the different time sampling resolution scenarios. Our near-term future work includes an analysis of the interaction between offshore wind and wave energy and LDS, and we intend to run scenarios with various temporal resolutions, including a scenario with hourly resolution overall 365 days.

## Reporting summary

Further information on research design is available in the Nature Portfolio Reporting Summary linked to this article.

## Data availability

The SWITCH 2.0 output data generated in this study has been deposited in the dataset supporting "Offshore Wind and Wave Energy Can Reduce Total Installed Capacity Required in Zero Emissions Grids" Figshare database under ref. 80. The results data generated in this study are provided in the Supplementary Information/Source Data file. Source data are provided with this paper.

## Code availability

The SWITCH WECC model (version v2.0.0) is open source and published on GitHub (https://github.com/REAM-lab/switch). The source code for the SWITCH 2.0 model used in this study has also been deposited in Zenodo under [https://doi.org/10.5281/zenodo.11116972][65].

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

## Acknowledgements

The authors would like to thank the U.S. Department of Energy for their funding through the Water Power Technologies Office (award number DE-EE0009443) (D.A.S., P.H.G., and D.M.K.). We acknowledge the support of Marcus Lehmann from CalWave for his contributions to the grant proposal writing. We acknowledge the early involvement in the project of Shiny Choudhury for downloading preliminary 14 coordinates with time series to calculate wave energy capacity factors and test preliminary SWITCH WECC toy runs for the 2022 Q1 U.S. Department of Energy quarterly report. We would like to thank Sarah Kurtz for her support and feedback on the newly imported offshore wind and wave energy candidate projects by N.G. and P.S.P. We would also like to thank Matthias Fripp, Josiah Johnston, Rodrigo Henriquez-Auba, Benjamin Maluenda, Ana Mileva and Jimmy Nelson for their prior contributions and developments of the SWITCH model. Special thanks go to Matthias Fripp and Rodrigo Henriquez-Auba for their courtesy of sharing the mathematical formulation Latex files of their Supplementary Information for us to continue expanding upon. This material is based upon work supported by the U.S. National Science Foundation Graduate Research Fellowship Program under Grant No. DGE-2038238 (N.G.). Any opinions, findings, conclusions, or recommendations expressed in this material are those of the author(s) and do not necessarily reflect the views of the U.S. National Science Foundation.

## Author contributions

N.G. extracted 8811 coordinates with time series from NREL's WPTO West Coast sites to calculate hourly capacity factors for the 101 sites of industry interest selected by R.D. P.S.T., and N.G. collaborated on data extraction, cleaning, and calculations of hourly capacity factors. P.S.P. extracted coordinates with time series for offshore wind sites (collocated) from NREL Offshore NW Pacific Dataset and calculated hourly capacity factors. N.G. and P.H.G. designed cost scenarios for this study. N.G. and P.H.G. imported new data sets to the database and created new scenarios. N.G. and P.S.T. ran SWITCH WECC under all scenarios, analyzed output data, created figures, tables, and wrote the manuscript with the guidance of P.H.G. R.W. supported N.G. in the creation of figures and tables for the Supplementary Information. B.M. provided technical support and resources to calculate hourly capacity factors for wave energy sites. SWITCH WECC development: (since 2020) M.S., (since 2020) P.S.P., (since 2020) J.S., and (since 2016) P.H.G. have been developing SWITCH WECC to this stage. P.H.G. and D.A.S. conceptualized the study. P.H.G., D.A.S., and D.A.K. acquired funds. P.H.G. supervised and advised the direction of this study, how to present findings, and edited the manuscript. N.G., P.S.T., B.M., J.S., R.W., D.M.K., D.A.S., and P.H.G. reviewed and edited the manuscript.

## Competing interests

Ryan Davidson is an employee at CalWave, a wave energy company, however the WEC we use in our study (RM6 from the National Renewable Energy Laboratory) has no similarities with the private designs at CalWave. Ryan Davidson's expertize supports the methodology and choice of sites of industry interest for this study. The remaining authors declare no competing interests.
