## [Peer Review File · Nature Communications]

Offshore wind and wave energy can reduce total installed capacity required in zero emissions gridsREVIEWER COMMENTS

Reviewer #1 (Remarks to the Author):

This paper has identified various combinations of cost targets for offshore wind and wave energy and has investigated their impact on total installed capacity and curtailment within a future zero-emission grid using a power system planning model. The authors have presented a substantial amount of data and detailed results, and the manuscript is well-structured and easily comprehensible.

While most of the analysis results are logically sound and provide several foreseeable implications, I believe that the manuscript could be improved based on the following comments:

1. The manuscript and the adopted planning model seem to offer sensitivity analyses for different cost portfolios of offshore wind and wave energy. The key findings appear to revolve around the impacts of changes in investment costs for these two generation technologies on the system's capacity, costs, and curtailment (whether it decreases or increases). It would be beneficial to emphasize the novel aspects of this model in comparison to previous studies.
 2. The authors mention that they employ the well-known long-term capacity expansion model, SWITCH, to model the zero-emission grid. However, they appear to plan the selected grid for a target year, specifically 2050. Considering a dynamic planning method that spans multiple years or stages could more effectively illustrate the effects of capital cost variations in generation technologies on capacity and generation mixes. It would be helpful if the authors could explain their rationale for choosing the target year planning approach over multi-year planning.
 3. Given that the authors are planning for a specific target year, using a 4-hour interval dispatch instead of hourly simulation may introduce potential inaccuracies in estimating operational results. Providing a rationale for this choice or discussing its limitations would be valuable.
 4. Some studies, such as [1], take into account offshore wind supply curves, which indicate the maximum available wind capacity at different LCOE levels for coastal zones, within their planning models. It would be advisable for the authors to discuss and compare their cost targeting methodology with that presented in [1] to highlight the differences and similarities.
 5. It would be beneficial for the authors to elaborate on how they account for power generation losses, wind variations, and capacity factor reductions resulting from wind farm wakes within their planning model. Providing clarity on this aspect can enhance the comprehensiveness of the study.
- [1] Guo, X., Chen, X., Chen, X. et al. Grid integration feasibility and investment planning of offshore wind power under carbon-neutral transition in China. *Nat Commun* 14, 2447 (2023). <https://doi.org/10.1038/s41467-023-37536-3>.

Reviewer #2 (Remarks to the Author):

The subheadings are not in heading format. they should be revised.

There are references to additional material in many places, but very detailed information is given about what is given in the additional material. It is not summarized. This disrupts fluency.

The writing style is not fluent. the reader gets bored. everything and every value is expressed one by one, whereas the results could have been written in a fluent language without falling into repetition.

There are repetitive sections. Repetitive paragraphs under different headings should be simplified.

Reviewer #3 (Remarks to the Author):

The paper focuses on the investigation of the system-wide impacts of integrating offshore wind (both fixed-bottom and floating) and wave energy into a carbon free electricity mix, using a least-cost capacity planning model with high spatial and temporal resolution, detailed power systems modeling, and a wide variety of candidate technologies. The authors present useful findings and insights in the Sections 2 (Results) and 3 (Discussion), that derived from their analysis. The key

contributions of this work are clear. The work presented and its added value are well articulated and stated in the manuscript. However, the work needs further revision in some parts of the manuscript and mainly in the "Methods" Section:

(1) The methodology is well articulated and stated in the manuscript, however a schematic depiction of the proposed methodological framework is still missing in the main text. It should be added in the "Methods" Section, since the reader is easily lost without such a scheme and it could help the reader to understand the contributions of the present work. Such a scheme is mandatory for studies, which are based on specific steps or/and stages and these steps/stages should be followed in order to derive accurate results. It is also very beneficial for the repeatability of the study.

(2) Lines 866, 906, etc., the authors refer that excluded from their site-selection analysis "restricted areas". Please mention the specific areas that have been included in that term, such as verified shipping routes, undersea archeological or historical sites, touristic sites, underwater cables and pipelines, aquaculture zones (non-protected), areas of landscape value, etc. The authors should mention the types of the areas that have been excluded and if they didn't consider certain exclusion criteria, due to the lack of GIS data, they should mention it and describe the potential impacts of this limitation on the final results.

(3) Lines 914-928, did the authors exclude areas based on the water depth? For example, did they exclude areas that have water depth larger than 500 m or 300 m, due to technical limitations? They should mention the water depth limitations that have been applied to their study. Also, they have to mention the water depth ranges of the candidate project areas. The technical and economic requirements of offshore projects are changing based on the water depth.

(4) Are there national policies or spatial planning regulations, that have been included in their analysis? For example, offshore wind and wave projects should be installed at least 4 km far away from archaeological sites or 3 km from residential areas or they should be inside of the national territorial waters? All the proposed projects' sites are inside of the national territorial waters or are only inside of the Exclusive Economic Zone? In general, are there national spatial planning regulations for RES that restricted the projects' availability/suitability?

(5) In Line 860, the authors mention that they rank the sites with high offshore wind and wave energy potential. In Section 6.2.1, they refer the parameters that they used for this assessment process. However, it is not clear the method used for this evaluation. Did they use a multicriteria decision-making (MCDM) method for their prioritization or did they follow a simpler approach? Could you please describe more that scoring framework developed by the CalWave? Also, they assigned weights to their parameters. The weights affect strongly the final prioritization results. If you change the weights, you can have completely different prioritization results. They should describe the reasons that they assigned higher weights to some parameters and lower weights to others.

(6) (minor) Please include the abbreviation of "marine protected areas" in Line 864 and please delete it in Lines 896-897. In Line 1204, please erase the word "photovoltaic" and keep only the abbreviation, you have already mentioned this abbreviation in Line 197. In Line 1023, please erase the words "wave energy converters" and use the abbreviation "WECs" that you define in Line 732. Please check your abbreviations in the whole manuscript, you may find a few more revisions that they should be made.

Response to Reviewers

Submission NCOMMS-23-43388

Dec 2023

Natalia Gonzalez, Paul Serna-Torre, Pedro Sanchez-Perez, Ryan Davidson,
Bryan Murray, Martin Staadecker, Julia Szinai, Rachel Wei, Daniel M.
Kammen, Deborah Sunter, and Patricia Hidalgo-Gonzalez

We would like to thank the reviewers for their valuable feedback in this review round. In what follows, we provide detailed responses to their comments.

Response to Reviewer 1

This paper has identified various combinations of cost targets for offshore wind and wave energy and has investigated their impact on total installed capacity and curtailment within a future zero-emission grid using a power system planning model. The authors have presented a substantial amount of data and detailed results, and the manuscript is well-structured and easily comprehensible.

While most of the analysis results are logically sound and provide several foreseeable implications, I believe that the manuscript could be improved based on the following comments:

1. *The manuscript and the adopted planning model seem to offer sensitivity analyses for different cost portfolios of offshore wind and wave energy. The key findings appear to revolve around the impacts of changes in investment costs for these two generation technologies on the system's capacity, costs, and curtailment (whether it decreases or increases). It would be beneficial to emphasize the novel aspects of this model in comparison to previous studies.*

We thank the reviewer for this recommendation as we agree it is of high importance to make this clear to the reader. In lines 149-156, we have identified gaps in the literature and other models as follows, “the large body of work analyzing the potential electricity generation mixes for a future decarbonized grid and the several studies investigating certain impacts of integrating offshore wind and/or wave energy” fall short of the following things:

- including both fixed-bottom and floating offshore wind
- including wave energy in the mix of candidate renewable technologies
- understanding the technical implications of their relative deployment

In lines 156-163, we state that this study is the first to “investigate the system-wide impacts of integrating offshore wind (both fixed-bottom and floating) and wave energy into a carbon-free electricity mix using a least-cost capacity planning model with high spatial-resolution, detailed power systems modeling, and a wide variety of candidate technologies.” Please see our response to comment 3 below for more information regarding the temporal resolution.

Furthermore, lines 233-245 state the main contributions of this work as:

1. modeling offshore wind and wave energy as independent technologies with the possibility of collocation in a power system capacity expansion model of the Western Interconnection,
2. identifying, for the first time, cost targets for offshore wind and wave energy to become cost effective in a zero emissions grid,

3. observing a 17% of reduction in total installed capacity by 2050 when offshore wind and wave energy are fully deployed, and
 4. quantifying how lower wave energy cost targets result in lower total transmission expansion, and on the other hand, lower offshore wind cost targets result in higher transmission expansion.
2. *The authors mention that they employ the well-known long-term capacity expansion model, SWITCH, to model the zero-emission grid. However, they appear to plan the selected grid for a target year, specifically 2050. Considering a dynamic planning method that spans multiple years or stages could more effectively illustrate the effects of capital cost variations in generation technologies on capacity and generation mixes. It would be helpful if the authors could explain their rationale for choosing the target year planning approach over multi-year planning.*

We thank the reviewer for this comment as it is an important aspect in capacity expansion planning. The scenarios we ran in SWITCH to obtain the presented results included investment periods 2020, 2030, 2040, and 2050, but the presented results focused on the year 2050. The reason for this is because 2050 is the year that system-wide carbon emissions must reach 0 in our model. Since our research question seeks to understand how various offshore wind and wave energy cost targets affect specifically the cost-optimal zero-emissions grid, we center the discussion on the year 2050. Furthermore, since carbon emissions are allowed to various extents in investment periods 2020, 2030, and 2040, and hence fossil fuel generators are dispatched in those periods, we do not see the full effects caused by increasing penetrations of offshore wind and wave energy until the grid is fully comprised of carbon-zero resources. This is observable when comparing the results across all investment periods. We added the results for all investment periods to the supplemental materials so that readers may observe these trends. We also added a clarification to the introduction of the main text that specifies which investment periods were simulated in our study and where in the supplemental materials they may find results for investment periods before 2050. Thank you for helping us strengthen the work and inspiring us to include these results in the supplemental materials.

3. *Given that the authors are planning for a specific target year, using a 4-hour interval dispatch instead of hourly simulation may introduce potential inaccuracies in estimating operational results. Providing a rationale for this choice or discussing its limitations would be valuable.*

We thank the reviewer for this suggestion. We added a clarification in the methods section of the manuscript to explain our reasoning for using a four-hour interval dispatch. We state, “the reduced complexity from using a four-hour time interval allows us to spend more computational effort on having a high geographical resolution for potential sites and having 2030, 2040 and 2050 investment decisions to better understand the transition. A faster run time from sampling hours also allowed us to create many scenarios to evaluate the relative deployment of offshore wind and wave energy. We included a discussion of the temporal resolution in the limitations section of the paper. In our previous study that uses SWITCH to evaluate the impact of using various time sampling resolutions on the utilization of long-duration energy storage (LDES) [1], we find that although the utilization of LDES is affected by the time sampling resolution used, the overall installed capacity mix does not vary largely between the different time sampling resolution scenarios. Our near-term future work includes an analysis on the interaction between offshore wind and wave energy and LDES, and we intend to run scenarios with various temporal resolutions, including a scenario with hourly resolution over all 365 days.”

4. *Some studies, such as [2], take into account offshore wind supply curves, which indicate the maximum available wind capacity at different LCOE levels for coastal zones, within their planning models. It would be advisable for the authors to discuss and compare their cost targeting methodology with that presented in [2] to highlight the differences and similarities.*

We thank the reviewer for bringing up this point of discussion related to capacity expansion modeling and how to represent sites for renewable sources with time-varying capacity factors while being mindful of computational complexity. In our modeling we opted to “spend” computational complexity by using a disaggregated approach where we do not cluster regions based on LCOE to represent wind/solar/wave energy farms. We considered this aspect important for this study because the shapes of the time-varying capacity factors could play an important role when considering offshore energy and how it may complement onshore energy and load. Naturally, we did aggregate the raw data, but only based on geographical proximity and similarity of the hourly capacity factors time series for a year. This representation allows the optimization problem to freely choose what is optimal from a physical perspective (which shapes of capacity factors better complement each other and the expected loads) and also from an economic perspective (minimizing costs). When sites are clustered by LCOE, a compromise is being made on the representation of the hourly capacity factors for each clustered site. Hence, a renewable energy cluster based on the LCOE method, may aggregate sites that are not necessarily similar with respect to their hourly dispatch time series, but they have a very similar LCOE. LCOE clustering does not take into account that the optimization model can choose to build a site even if in some hours it would be optimal to curtail, hence the site would result in a lower LCOE than what was assumed by the cluster. Given that the focus of our study was on renewable’s siting, their economics and the shape of their hourly capacity factors, we decided to not aggregate by LCOE and let the model choose which sites to build.

Coincidentally, we are working on another project where we are quantifying what’s the impact on investment and operations in capacity expansion planning as a function of aggregating renewable sites by LCOE with higher or lower geographical resolution (to avoid the aforementioned issue). We hope the findings of our upcoming study will be insightful for the community to better understand the impacts of clustering by LCOE.

5. *It would be beneficial for the authors to elaborate on how they account for power generation losses, wind variations, and capacity factor reductions resulting from wind farm wakes within their planning model. Providing clarity on this aspect can enhance the comprehensiveness of the study.*

We thank the reviewer for suggesting ways to strengthen the study. We have incorporated these suggestions. In section 5.3 of the supplemental materials, we describe in detail how existing and candidate transmission lines are modeled in SWITCH. Regarding efficiency losses, SWITCH assumes a 1% efficiency loss for every 100 miles of distance, which is based on typical losses for high-voltage transmission [3]. Wind variations are captured in the capacity factor time series associated to each onshore and offshore wind project in SWITCH, which describe the ratio of power output to rated power of each generator at every hour of the year. Offshore wind data used to determine capacity factors for the representative offshore wind turbine at each candidate project area was collected from the National Renewable Energy Lab (NREL) Offshore NW Pacific Dataset [4]. Wave characteristic data used to determine capacity factors for the representative wave energy converter (WEC) at each candidate project area was collected from the U.S. Department of Energy (DOE) Water Power Technology Office’s (WPTO) U.S. Wave dataset [5]. We have included the description of this process in section 6. All variable renewable energy generators that are input as candidate projects in SWITCH have an associated capacity

factor time series based on various wind, solar, and wave data sets, as described in section 5.2 of the supplemental materials. We do not consider wake effects in the offshore wind or wave energy capacity factors. As recommended by the reviewer, we added clarity on this aspect to the main text. We add the statement, “We do not consider wake effects of WECs because there is limited information on this topic, and wake effects can vary largely from one WEC design to another.” Regarding offshore wind wake effects, we add the statement, “Furthermore, we do not consider wake effects of offshore wind turbines given that this is a variable dependant on specific farm array design that can be minimized by developers through strategic design.”

Response to Reviewer 2

1. *The subheadings are not in heading format. they should be revised.*

Thank you for pointing this out. We changed all subheadings to bold, non-italicized fonts with 60 characters or less, as required by the re-submission formatting guidelines.

2. *There are references to additional material in many places, but very detailed information is given about what is given in the additional material. It is not summarized. This disrupts fluency.*

Thank you for your comment. Rather than referencing the supplemental materials after each individual result, we summarize the relevant tables, figures, and analyses that can be referenced in the supplemental materials at the end of each subsection. There are now significantly fewer references to the supplemental materials in the main text. We believe now the text is smoother thanks to this recommendation.

3. *The writing style is not fluent. the reader gets bored. everything and every value is expressed one by one, whereas the results could have been written in a fluent language without falling into repetition.*

Thank you for your comment. We agree that the original submission had many numerical details that disrupted the flow of the writing. To address this, we modified the results section to report only trends, maximum increases/decreases in each metric across the scenarios, and the technical reasoning for each result. We also cut repetitive references to the scenario numbers and supplemental materials. Furthermore, we added a graphic that summarizes the methodology so that the main steps of our study are made clear in an easily digestible fashion. We appreciate the feedback as a clear communication style is important to convey the scientific findings.

4. *There are repetitive sections. Repetitive paragraphs under different headings should be simplified.*

Thank you for pointing out these redundancies. We agree that redundant material in separate sections should be simplified. To address this, we removed the redundancies between the introduction and methods sections and added short references to previously stated materials where necessary.

Response to Reviewer 3

The paper focuses on the investigation of the system-wide impacts of integrating offshore wind (both fixed-bottom and floating) and wave energy into a carbon free electricity mix, using a least-cost capacity planning model with high spatial and temporal resolution, detailed power systems modeling, and a wide variety of candidate technologies. The authors present useful findings and insights in the Sections 2 (Results) and 3 (Discussion), that derived from their analysis. The key contributions of this work are clear. The work presented and its added value are well articulated and stated in the manuscript. However, the work needs further revision in some parts of the manuscript and mainly in the “Methods” Section:

1. The methodology is well articulated and stated in the manuscript, however a schematic depiction of the proposed methodological framework is still missing in the main text. It should be added in the “Methods” Section, since the reader is easily lost without such a scheme and it could help the reader to understand the contributions of the present work. Such a scheme is mandatory for studies, which are based on specific steps or/and stages and these steps/stages should be followed in order to derive accurate results. It is also very beneficial for the repeatability of the study.

Thank you for your helpful comment. As the reviewer suggested, we added a schematic at the beginning of the methodology section that summarizes the main steps of methodological framework. We include the figure (Fig. 1) below for reference.

Figure 1: Methodology flow chart.

2. Lines 866, 906, etc., the authors refer that excluded from their site-selection analysis “restricted areas”. Please mention the specific areas that have been included in that term, such as verified

shipping routes, undersea archeological or historical sites, touristic sites, underwater cables and pipelines, aquaculture zones (non-protected), areas of landscape value, etc. The authors should mention the types of the areas that have been excluded and if they didn't consider certain exclusion criteria, due to the lack of GIS data, they should mention it and describe the potential impacts of this limitation on the final results.

Thank you for your comment to help us clarify what were the areas excluded and associated limitations. By “restricted areas”, we mean restricted military activity areas that are closed to public access. We clarified this ambiguous statement by changing “military danger zones and restricted areas” to “military danger zones and restricted military activity areas.” In lines 759-768, we clarify the 3 strictest classifications of marine protected areas that are used to exclude some sites of industry interest, which are “No Take,” “No Impact,” and “No Access” zones. We added the following line to this paragraph to clarify that the candidate project areas do not overlap with Heritage Marine Sites: “No United Nations Educational, Scientific and Cultural Organization (UNESCO) World Heritage Marine Sites (WHMSs) overlap with any of the sites of industry interest [6].” The UNESCO WHMSs are sites that are considered to have “outstanding universal value” and meet at least one of UNESCO’s 6 cultural and 4 natural criteria [6]. There is limited data availability for archaeological or historical sites that are not included in UNESCO’s WHMSs. We do not include shipping routes as a filtering parameter because they cover almost the entire Pacific coastline [7] and are expected to adjust to marine energy farms in the future. Figure 2 below shows the extent to which commercial shipping routes cover the U.S. West Coast. However, we agree it would be relevant to exclude routes that will be identified as essential for economic development in the next decades. We also do not include existing underwater cables or pipelines in the filtering criteria because it is expected that offshore wind and wave energy transmission lines will have to cross existing underwater assets, and that safe asset crossing procedures will need to be designed and approved [8]. Furthermore, we added a detailed paragraph in section 1.1 of the supplemental materials that describes what parameters are not considered when filtering the sites of industry interest, why they are not considered, and what we believe the potential impact from omitting these parameters from the filtering process could be on the results. We have also included a new “Limitations” section under “Methods” to discuss how results can change based on the areas that were and were not excluded.

3. *Lines 914-928, did the authors exclude areas based on the water depth? For example, did they exclude areas that have water depth larger than 500 m or 300 m, due to technical limitations? They should mention the water depth limitations that have been applied to their study. Also, they have to mention the water depth ranges of the candidate project areas. The technical and economic requirements of offshore projects are changing based on the water depth.*

We do not enforce a maximum water depth on the offshore wind and wave energy candidate projects because it is uncertain what water depths will be possible to install marine energy devices in the year 2050 due to technological advancements over the coming decades. Furthermore, the official Offshore Wind Call Areas that represent offshore wind energy sites of interest as identified by BOEM are between 200m and 1300m deep. However, only 5 of the 101 candidate project areas have any portions of their areas beyond the 1300m depth contour. The water depth data from [9], as shown in Figure 3 shows contours up to 1500m depths. There we can observe the 5 sites from the 101 candidate project areas that have portions of their areas beyond the 1500m depth contour. The study in [10] shows deep water floating offshore wind turbines in 600m to 900m depths. Only 8 of the 101 candidate project areas have any portions of their areas beyond the 900m depth contour and as such, this is reflected in their

Figure 2: 2004-2005 shipping traffic density map [7].

cost assumption and design. We have added this explanation of water depths to section 1.1 of the supplemental materials.

4. *Are there national policies or spatial planning regulations, that have been included in their analysis? For example, offshore wind and wave projects should be installed at least 4 km far away from archaeological sites or 3 km from residential areas or they should be inside of the national territorial waters? All the proposed projects' sites are inside of the national territorial waters or are only inside of the Exclusive Economic Zone? In general, are there national spatial planning regulations for RES that restricted the projects' availability/suitability?*

Thank you for this comment. These are all very important factors to take into consideration. Hence, we reviewed in detail the Offshore Wind Energy Development Legal Framework by the Congressional Research Service [11] and the Information Guidelines for a Renewable Energy Construction and Operations Plan (COP) by BOEM [12]. Although there are many stringent requirements regarding what offshore energy developers must include in their COPs and regarding the federal and state permitting processes, we found no national policies or spatial planning regulations that explicitly restrict offshore wind or wave energy development in terms of distance from specific areas or assets. Hence, we exclude the consideration of distance from residential areas in the filtering process for the sites of industry interest. As mentioned above in our response to comment 2 from reviewer 3, we did ensure that none of our candidate project areas overlap with UNESCO's WHMSs.

Regarding the U.S. jurisdiction over the ocean, all offshore wind and wave energy candidate projects fall within the Exclusive Economic Zone (EEZ). The National Oceanic and Atmospheric Administration (NOAA) states that the EEZ is the zone in which the U.S. has "sovereign rights for the purpose of exploring, exploiting, conserving and managing natural resources, whether living and nonliving, of the seabed and subsoil and the superjacent waters and with regard to other activities for the economic exploitation and exploration of the zone, such as

Figure 3: U.S. West Coast ocean depth contours from 200 to 1500 meters overlaid on candidate project areas [9]. The legend on the left shows ocean depth in meters.

the production of energy from the water, currents and winds” [13]. Most of the offshore wind and wave energy candidate projects fall within the U.S. Contiguous Zone, and about 3/4 of the projects fall within the Territorial Seas. We added this clarifying information to section 1 of the supplemental materials.

5. *In Line 860, the authors mention that they rank the sites with high offshore wind and wave energy potential. In Section 6.2.1, they refer the parameters that they used for this assessment process. However, it is not clear the method used for this evaluation. Did they use a multi-criteria decision-making (MCDM) method for their prioritization or did they follow a simpler approach? Could you please describe more that scoring framework developed by the CalWave? Also, they assigned weights to their parameters. The weights affect strongly the final prioritization results. If you change the weights, you can have completely different prioritization results. They should describe the reasons that they assigned higher weights to some parameters and lower weights to others.*

Thank you for your question. To clarify, we do not rank the 100 industry sites of interest to give preference in SWITCH to any of the sites identified through CalWave’s proprietary scoring framework. We use most of the 100 sites identified by Calwave as the industry sites of interest to generate the candidate project areas for offshore wind and wave energy, and SWITCH decides which sites to build wave energy and/or offshore wind energy capacity based on the least-cost optimization subject to the model’s constraints (e.g., serving sampled hourly loads being mindful of hourly capacity factors by lat/lon, etc.). CalWave uses NREL’s report

Figure 4: The U.S. EEZ (purple), Contiguous Zone (green) and Territorial Sea (red) boundaries with respect to the offshore wind and wave energy candidate project areas [13].

on Marine Hydrokinetic Energy Site Identification and Ranking [14] as a guideline for their own ranking framework. They use the scoring and ranking framework to evaluate and narrow down several potential sites on the U.S. West Coast to the top 100 sites. It is important to note that there are some key differences between CalWave’s methodology and that of NREL. We added an explanation of these differences, as well as a list of the parameters in order of assigned weight from highest weight to lowest weight, to the methodology section of the manuscript.

6. *(minor) Typos with abbreviations:*

- a) *Please include the abbreviation of “marine protected areas” in Line 864 and please delete it in Lines 896-897.*
- b) *In Line 1204, please erase the word “photovoltaic” and keep only the abbreviation, you have already mentioned this abbreviation in Line 197.*
- c) *In Line 1023, please erase the words “wave energy converters” and use the abbreviation “WECs” that you define in Line 732. Please check your abbreviations in the whole manuscript, you may find a few more revisions that they should be made.*

We appreciate the reviewer for catching these abbreviation typos. We have corrected them and checked to ensure that there are not similar typos elsewhere in the text.

Note to All Reviewers

We are grateful to all reviewers for their valuable feedback. As we implemented the comments, we reviewed all of our process and results again to ensure that no details were overlooked. Upon doing so, we found that one small section of the initial results was inaccurate. The previous submission showed curtailment decreasing with decreased wave and offshore wind energy cost targets, when in reality, it increases. We found that system-wide curtailment increases with decreasing offshore wind and wave energy cost targets. The result, logic, and discussion have been updated to reflect this corrected calculation. We are happy to report that all other results have been checked and confirmed by our team.

References

- [1] P. A. Sánchez-Pérez, S. Kurtz, N. Gonzalez, M. Staadecker, P. Hidalgo-Gonzalez, Effect of time resolution on capacity expansion modeling to quantify value of long-duration energy storage, 2022 IEEE Electrical Energy Storage Application and Technologies Conference (EESAT) (2022) 1–5doi:10.1109/EESAT55007.2022.9998031.
- [2] X. Guo, X. Chen, X. Chen, P. Sherman, J. W. . M. McElroy, Grid integration feasibility and investment planning of offshore wind power under carbon-neutral transition in china, Nature Communications 14 (2023) 2447. doi:10.1038/s41467-023-37536-3.
- [3] M. Brown, W. Cole, K. Eurek, J. Becker, D. Bielen, I. Chernyakhovskiy, S. Cohen, A. Frazier, P. Gagnon, N. Gates, D. Greer, J. Ho, P. Jadun, T. Mai, M. Mowers, C. Murphy, A. Rose, A. Schleifer, D. Steinberg, Y. Sun, N. Vincent, E. Zhou, K. Lamb, M. Zwerling, Regional energy deployment system (reeds) model documentation: Version 2019, Tech. Rep. NREL/TP-6A20-74111, National Renewable Energy Laboratory (NREL) (2020). doi:10.2172/1606151.
- [4] N. N. R. E. Laboratory), Offshore nw pacific wind data download, <https://developer.nrel.gov/docs/wind/wind-toolkit/offshore-nw-pacific-download/> (2020).
- [5] N. N. R. E. Laboratory), Doe water power technology office’s (wpto) us wave dataset, https://github.com/NREL/hsds-examples/blob/master/datasets/US_Wave.md (May 2021).
- [6] S. United Nations Educational, C. O. (UNESCO), Boundaries of unesco world heritage marine sites (v02), <https://doi.org/10.14284/592> (2023).
- [7] R. Watch, Commercial shipping lanes, <https://resourcewatch.org/data/explore/com012-Global-Shipping-Lanes> (2005).
- [8] N. Y. S. E. Research, D. A. (NYSERDA), Offshore wind submarine cabling overview, <https://databasin.org/datasets/60f4698c750a48b5ba2bcd6808fd9388/> (2021).
- [9] B. of Ocean Energy Management, 100m depth contours, <https://doi.org/10.14284/592> (2017).
- [10] N. R. E. L. (NREL), Loads analysis of a floating offshore wind turbine using fully coupled simulation, https://digital.library.unt.edu/ark:/67531/metadc891165/m2/1/high_res_d/909454.pdf (2007).
- [11] U. S. D. of the Interior, O. of Renewable Energy Programs, B. of Ocean Energy Management, Information guidelines for a renewable energy construction and operations plan (cop), <https://sgp.fas.org/crs/misc/R40175.pdf> (2020).

- [12] C. R. Service, Offshore wind energy development: Legal framework, r40175, <https://sgp.fas.org/crs/misc/R40175.pdf> (2005).
- [13] N. Oceanic, A. A. (NOAA), U.s. maritime limits & boundaries, <https://nauticalcharts.noaa.gov/data/us-maritime-limits-and-boundaries.html>.
- [14] Marine hydrokinetic energy site identification and ranking methodology part i: Wave energy,” by levi kilcher and robert thresher, Tech. rep., National Renewable Energy Laboratory (NREL) (2016).

REVIEWERS' COMMENTS

Reviewer #1 (Remarks to the Author):

The authors have well responded to the reviewer's comments. I look forward to seeing more of the authors' upcoming work.

Reviewer #1 (Remarks on code availability):

The authors have included detailed code and manuals in the GitHub repository and provided installation and usage steps for the software and code in the README file. The directory structure is very clear with strong readability.

Reviewer #2 (Remarks to the Author):

The authors focused on all comments so I recommend that the article can be accepted for publication.

Reviewer #3 (Remarks to the Author):

The authors addressed my comments and conducted a productive revision of their manuscript. The added value of their work is well articulated and stated in the revised manuscript.

Reviewer #3 (Remarks on code availability):

I'm not able to assess this. The code provides a README file and several other necessary instructions and files (e.g., code and instructions for developing SWITCH modules) for installing and running the model, step by step. Yes, it is a usable resource for the community, in order to be able to rerun the SWITCH electricity planning model adapted from the REAM research lab. However, it is required to install, run the different components of SWITCH, create SWITCH modules from scratch, connect to REAM database and modify it and implement several other relevant tasks, in order to be able to assess the degree of usability of the code and the degree of reproducibility of the results of the paper. All the above cannot be assessed and addressed during the review time of the paper.

Response to Reviewers

Submission NCOMMS-23-43388

May 2024

Natalia Gonzalez, Paul Serna-Torre, Pedro Sanchez-Perez, Ryan Davidson,
Bryan Murray, Martin Staadecker, Julia Szinai, Rachel Wei, Daniel M.
Kammen, Deborah Sunter, and Patricia Hidalgo-Gonzalez

We would like to thank the reviewers for their valuable feedback in this review round. In what follows, we provide detailed responses to their comments.

Response to Reviewer 1

The authors have included detailed code and manuals in the GitHub repository and provided installation and usage steps for the software and code in the README file. The directory structure is very clear with strong readability.

We thank the reviewer for his comments. The GitHub repository now also has a DOI associated with the version used in this study.

Response to Reviewer 2

The authors focused on all comments so I recommend that the article can be accepted for publication.

We thank the reviewer for this comment and we are happy to hear that you feel we addressed all your previous comments.

Response to Reviewer 3

The authors addressed my comments and conducted a productive revision of their manuscript. The added value of their work is well articulated and stated in the revised manuscript.

Remarks on code availability:

I'm not able to assess this. The code provides a README file and several other necessary instructions and files (e.g., code and instructions for developing SWITCH modules) for installing and running the model, step by step. Yes, it is a usable resource for the community, in order to be able to rerun the SWITCH electricity planning model adapted from the REAM research lab. However, it is required to install, run the different components of SWITCH, create SWITCH modules from scratch, connect to REAM database and modify it and implement several other relevant tasks, in order to be able to assess the degree of usability of the code and the degree of reproducibility of the results of the paper. All the above cannot be assessed and addressed during the review time of the paper.

We thank the reviewer for this comment. To address the code and data availability, we created a permanent DOI for the version of the source code used in this study, we made all the output files for the model available, and we provided all the source data in an Excel Workbook.